# Newcastle disease burden in Nepal and efficacy of Tablet I2 vaccine in commercial and backyard poultry production

**Rajindra Napit**[1,2◉], **Ajit Poudel**[1,2◉], **Saman M. Pradhan**[1,2◉], **Prajwol Manandhar**[2],
**Sajani Ghaju**[1], **Ajay N. Sharma**[1,2], **Jyotsna Joshi**[1,2], **Suprim Tha**[1], **Kavya Dhital**[1],
**Udaya Rajbhandari**[1], **Amit Basnet**[1], **Jessica S. Schwind**[3], **Rajesh M. Rajbhandari**[1,2], **Dibesh B. Karmacharya**[1,2,4] *

**1** Biovac Nepal Pvt. Ltd, Banepa, Kavre, Nepal, **2** Center for Molecular Dynamics Nepal, Kathmandu, Nepal,
**3** Institute for Health Logistics & Analytics, Georgia Southern University, Statesboro, Georgia, United States
of America, **4** Department of Biological Sciences, University of Queensland, Queensland, Australia

◉ These authors contributed equally to this work.
* dibesh@biovacnepal.com

Faculty of Veterinary Medicine, EGYPT

**Data Availability Statement:** All relevant data are
within the paper and its Supporting information
files.

## Abstract

Poultry (*Gallus domesticus*) farming plays an important role as an income generating enterprise in a developing country like Nepal, contributing more than 4% to the national Gross Domestic Product (GDP). Newcastle Disease (ND) is a major poultry disease affecting both commercial and backyard poultry production worldwide. There were more than 90 reported ND outbreaks in Nepal in 2018 with over 74,986 birds being affected. ND is responsible for over 7% of total poultry mortality in the country. Recent outbreaks of ND in 2021 affected many farms throughout Nepal and caused massive loss in poultry production. ND is caused by a single-stranded ribonucleic acid (RNA) virus that presents very similar clinical symptoms as Influenza A (commonly known as bird flu) adding much complexity to clinical disease identification and intervention. We conducted a nationwide ND and Influenza A (IA) prevalence study, collecting samples from representative commercial and backyard poultry farms from across the major poultry production hubs of Nepal. We used both serological and molecular assessments to determine disease exposure history and identification of strains of ND Virus (NDV). Of the 40 commercial farms tested, both NDV (n = 28, 70%) and IAV (n = 11, 27.5%) antibodies were detected in majority of the samples. In the backyard farms (n = 36), sero-prevalence of NDV and IAV were 17.5% (n = 7) and 7.5% (n = 3) respectively. Genotype II NDV was present in most of the commercial farms, which was likely due to live vaccine usage. We detected never reported Genotype I NDV in two backyard farm samples. Our investigation into 2021 ND outbreak implicated Genotype VII.2 NDV strain as the causative pathogen. Additionally, we developed a Tablet formulation of the thermostable I2-NDV vaccine (Ranigoldunga™) and assessed its efficacy on various (mixed) breeds of chicken (*Gallus domesticus*). Ranigoldunga™ demonstrated an overall efficacy >85% with a stability of 30 days at room temperature (25˚C). The intraocularly administered vaccine was highly effective in preventing ND, including Genotype VII.2 NDV strain.

**Funding:** Yes- Various components of our effort were supported by PSI (the Netherlands) and InnovationXchange grant from DFAT 60530159 (Australia)

**Competing interests:** The authors have declared that no competing interests exist.

# Background

Poultry (*Gallus domesticus*) farming is one of the major sources of protein and means of food security for growing population throughout the world [1]. Around 75 million broiler chickens are reared annually as a source of meat in Nepal, and the industry has rapidly grown in the recent years [2]. The present population of laying hens is more than 7 million, with more than 63 million eggs produced, and the meat produced from poultry exceeds over 17 thousand metric tons annually [3].

Newcastle Disease (ND) is one of the most devastating viral diseases of poultry globally [4]. In a developing country like Nepal, backyard poultry accounts for more than 45% of the total poultry production, and ND is one of the major diseases impacting this sector [5]. There were more than 90 reported ND outbreaks in Nepal in 2018 with over 74,986 birds being affected [6]. Overall ND is responsible for more than 7% of total poultry mortality in the country [7, 8].

ND is a highly infectious disease caused by a single-stranded RNA virus (avian orthoavula virus)- commonly known as Newcastle Disease Virus (NDV) [9]. The virus consists of an assembly of materials enclosed in a protein coat. This assembly is surrounded by an envelope, which is derived from the membranes of the host cell. Projecting from the envelope is a fringe of glycoprotein spikes. These are the haemagglutinin neuraminidase (HN) and fusion (F) glycoproteins that play important role in infection [10].

Although most avian species are susceptible to infection with NDV, chickens (*Gallus domesticus*) are most susceptible to the disease. ND is classified by the World Organization for Animal Health (WOAH) as a List 'A' disease of international concern. ND spreads rapidly, extends beyond national borders, and has serious socioeconomic consequences and major trade implications [11]. Depending on the NDV strain, infected birds can show a range of clinical signs. Age of bird, health and immune status of the host, presence of concurrent infections, and environmental conditions can influence the severity of the symptoms [12]. Some strains of ND cause no clinical signs, while others have high mortality rates [13]. Strains of NDV are divided into five groups or pathotypes based on clinical signs produced in experimentally-infected chickens [14].

The clinical representation caused by ND and Influenza A (IA) are often indiscernible [15]. And because some strains of Influenza A virus (IAV) (bird flu, e.g. H5N1) affect human health, early detection and intervention is crucial to discern between these two pathogens (NDV and IAV) for proper intervention strategies. NDV is limited to birds only whereas IAV has both human and animal health implications [16]. In chickens, clinical manifestations of ND include moderate to severe damage on the respiratory tract, often times leading to multi-organ failures. It can also decrease egg production, thereby severely impacting poultry farmers economically [17]. Outbreaks of ND or IA are often followed by embargoes and trading restrictions in affected areas [18].

Traditionally, disease diagnosis has relied on isolation and identification of the virus, which can take up to two weeks. Even with the advent of polymerase chain reaction (PCR) and real-time PCR (RT-PCR), sample collection and processing still take 1–2 days.

ND can be controlled and prevented using vaccines and by adapting a strict biosafety and biosecurity practiced [19]. Several ND vaccines are currently commercially available. Most of the ND vaccines deteriorate after one or two hours at room temperature, making them unsuitable for use in villages and on farms with limited cold chain facility. The I2-NDV vaccine was developed for local or regional production and used to control ND in places where cold chain is not reliable. The I2-NDV vaccine is a thermostable vaccine that remains effective for up to 30 days of storage at room temperature [20–22]. The high costs of maintaining cold chain for transportation and the use of other conventional vaccines makes ND vaccination impractical

and unsustainable in most rural areas. Having access to a locally-produced thermostable ND vaccine has a tremendous impact on preventing ND in a low resource country like Nepal.

Poultry vaccine production in Nepal has increased significantly (10–50% annually) in recent years [23]. There are three registered vaccine production laboratories in Nepal. F1 and ND R2B-based immunization started in 1968 in Nepal, and I2- NDV vaccine was introduced in 2008 [23]. There is limited information on various kinds of ND vaccine in use in Nepal and their efficacies in commercial and backyard poultry production [24, 25]. In one study, I2-NDV vaccine used in backyard chickens of Nepal showed a high antibody titer response at 14–30 day period, giving protective immunity for at least 3 months [25]. Although vaccine schedules vary, in general, I2- NDV live vaccines provide significant protection against NDV when administered at day 7 in broilers (45 day production cycle), and additional booster shots every 2–3 months for egg-laying chickens (18 month production cycle) [26].

In this study, we conducted a nationwide NDV and IAV prevalence study (2018–19) by collecting samples from representative commercial and backyard poultry farms from the major poultry production hubs of Nepal (Fig 1). This research included both serological and molecular assessments, which provided disease exposure, prior vaccination history and detection of various strains of NDV. Additionally, we developed a thermostable I2- NDV vaccine in a tablet formulation called Ranigoldunga™, which was found to be highly effective against NDV, including on a virulent NDV strain (Genotype VII.2) that caused nationwide outbreak in Nepal in 2021.

## Methods

Our study followed ethical guidelines as prescribed by the Department of Livestock Services (DLS) Nepal for the nationwide survey. Sampling and survey were conducted after obtaining written consent from the owners of commercial and backyard farms. Biological samples were collected using proper biosecurity measures in the presence of the farm owner or caretaker.

For vaccine tests, all *in vivo* experiments, animals were monitored daily by trained technicians and observational data were collected. Clinical signs of ND like greenish diarrhea, respiratory distress, decreased appetite, occurrence of ruffled feathers, lack of physical activity, and stunted growth, were observed and recorded [27]. We planned to euthanize birds that showed early evidence of severe suffering or distress (neck and leg paralysis) [28].

### Survey and sample collection of commercial and backyard poultry farms

In 2018 (June-December), we collected poultry samples from commercial (n = 40) and backyard (n = 36) farms from ten districts of Nepal. These districts are poultry production hubs that suffer from regular NDV and IAV outbreaks [29].

We collected samples (oral, cloacal and blood) from each commercial (n = 15 birds) and backyard farm (n = 3 birds) (Table 1). Birds were picked randomly for sample collection by walking along an imaginary diagonal line across the farm (Fig 1). Oral and cloacal swab samples were collected in viral transport media (VTM). Blood was collected in vaccutainer from the brachial vein, and serum was separated in cryovials by centrifuging blood (2000g for 10 minutes) by a licensed veterinarian. Samples were transported in liquid nitrogen tanks (-196˚C) to the lab in Kathmandu for further laboratory analysis.

**Biosecurity and biosafety risk assessment in poultry farms.** Biosecurity and biosafety practices survey of all the farms were conducted using a standardized questionnaire. The statistical analysis was done using IBM SPSS 20.

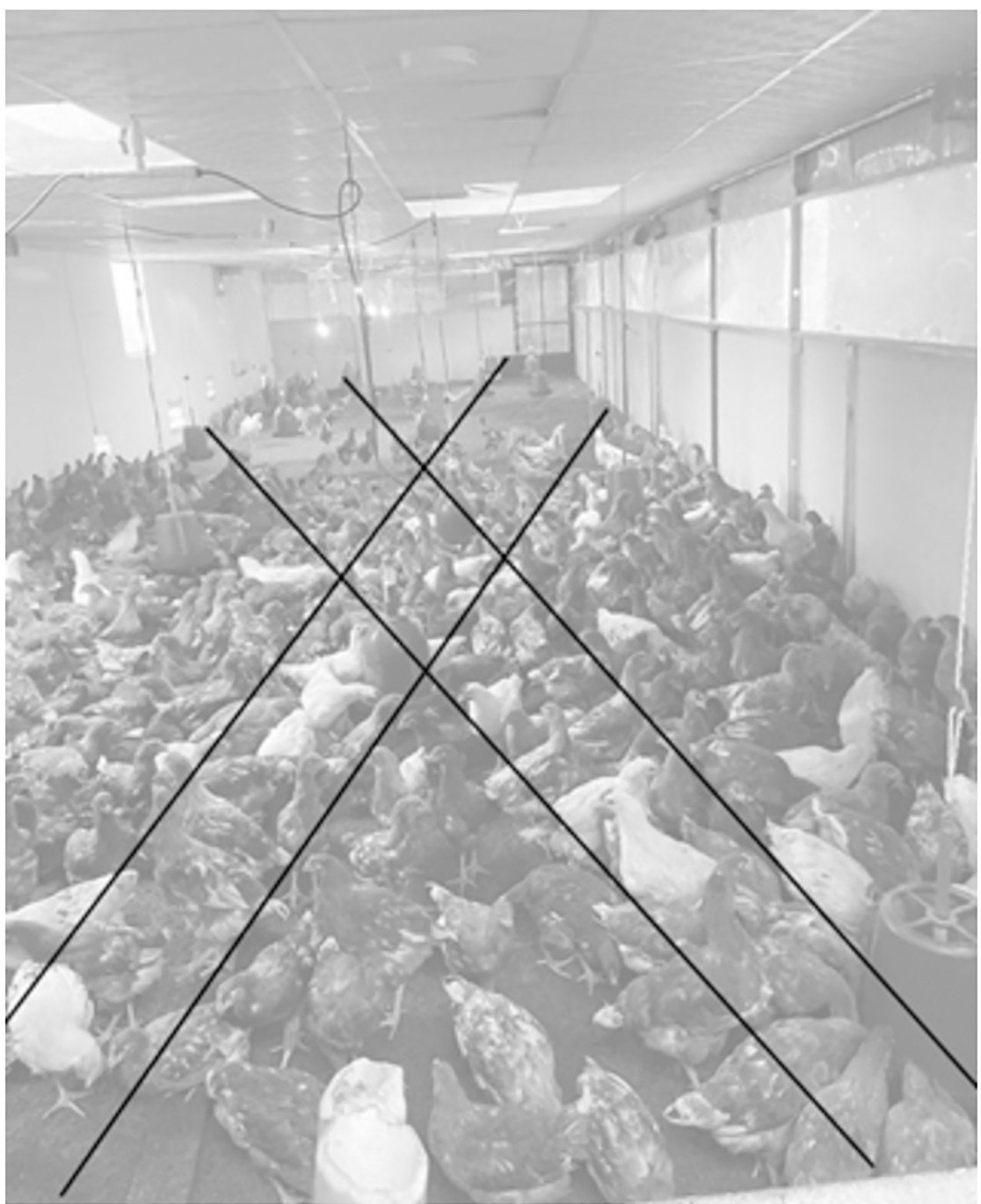

**Fig 1. Sampling strategy of chicken in a farm- an imaginary diagonal lines from where birds were randomly selected and sampled.**

**Table 1. Collected sample types from each commercial and backyard farm.**

|  | Total Number of Chickens | Oropharyngeal Swabs | Cloacal Swabs | Blood Specimens |
|---|---|---|---|---|
| Commercial farm | 15 | 15 | 15 | 15 |
| Backyard farm | 3 | 3 | 3 | 3 |

## Serological Assessment of NDV and IAV

Commercially available enzyme-linked immunosorbent assay (ELISA) kits (ID Screen® Influenza A Nucleoprotein and Newcastle Disease Nucleoprotein Indirect ELISA kits, France) were used for detecting and quantifying IAV and NDV antibodies (nucleoprotein) as per manufacturer's instructions. Five microliters of serum from each sample were used in each ELISA test.

## Molecular (PCR) assessment of NDV and IAV

**Sample processing strategy.** The oral, cloacal, and blood samples were collected from commercial (n = 40 farms, total birds = 600) and backyard (n = 36 farms, total birds = 108) poultry farms from across the major poultry production hubs of Nepal in 2018. Oropharyngeal swabs (n = 15 from each farm) and cloacal swabs (n = 15 from each farm) were pooled and collected in two separate single tubes. Altogether, 40 pooled oropharyngeal samples from commercial farms (n = 40) and 36 pooled oropharyngeal samples from backyard farms (n = 36) were collected.

**Viral RNA extraction and cDNA synthesis.** The pooled samples were vortexed for 2 minutes and spun down for 30 seconds. RNA from these samples containing VTM were extracted using Direct-zol RNA Miniprep Kits (Zymo Research, USA). 200ul of supernatant sample was added to 200ul of TRI Reagent® for lysis procedure. RNA extraction was done using the manufacturer's instructions.

cDNA was synthesized using iScript™ cDNA Synthesis Kit (BIORAD, California, USA). The reaction mix had 5x iScript™ reaction Mix (4μl), nuclease free water (10 μl), RNA template (4μl), and iScript reverse transcriptase enzyme (1μl), which was incubated initially at 25˚C for 5 mins, followed by 46˚C for 20 min. The reaction was inactivated at 95˚C for 1 min and eluted in a 50μl final volume.

**Detecting NDV and IAV using multiplex PCR assay.** A multiplex PCR assay was developed that can simultaneously detect both NDV and IAV in a single tube, thereby reducing screening cost and time. PCR primers were designed using reference sequence data from the NCBI database (S1 Table in S1 File). Sequence alignment was performed using MAFFT V. 7.0 [30] and Primer Blast [31].

Primers used for NDV are-

NDV ISO-F (`GCTCAATGTCACTATTGATGTGG`) and NDV ISO-R (`TAGCAGGCTGTCCCACTGC`)

Primers for IA are-

IAV ISO-F (`CTTCTAACCGAGGTCGAAACG`) and IAVM_ISO-R (`GGTGACAGGATTGGTCTTGTC`)

PCR conditions-

The reaction mixture consisted of nuclease free water (7μl), 2X Qiagen Master Mix (12.5μl), Taq Polymerase (1μl), 0.3 μM of primer and 3μl of cDNA. The samples were initially denatured at 95ºC for 10 minutes, followed by 45 cycles consisting of: 95ºC (denaturation) for 10 sec, 54ºC (annealing) for 15 sec, and 72ºC (extension) for 7 sec. The final extension was done at 72ºC for 5 minutes. The results were visualized in 1.5% Agarose gel.

## Characterization (Genotyping) of strains of NDV

**Amplification of Fusion (F) gene of NDV.** A tiled (juxtaposed multi-fragment) PCR and next generation sequencing assay was developed to analyze genotype (strain) of detected NDV (Fig 2). The entire Fusion (F) gene of NDV was segmented into four fragments (1–4) and amplified using PCR primers listed in Table 2. These PCR amplicons were purified using Montage Gel Extraction Kit (Merck, USA) and further processed for DNA sequencing.

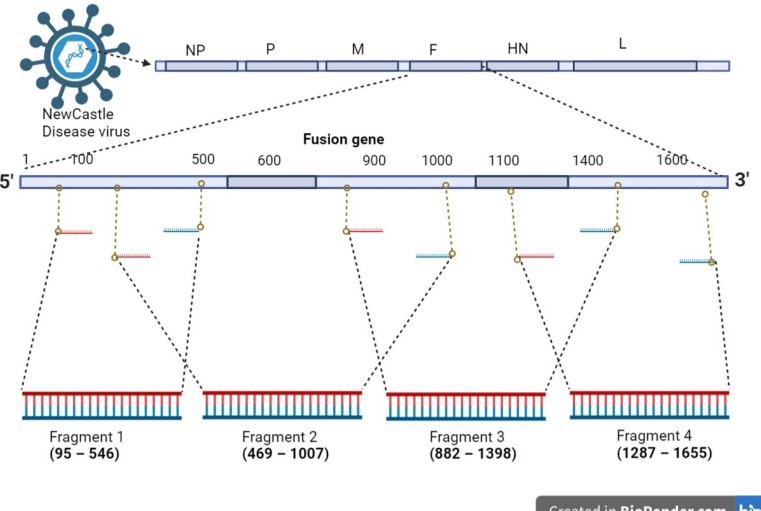

**Fig 2. Targeted Fusion gene tiled PCR amplification on NDV genome using designed primers.** The image was created using Biorender web app [32].

## Multiplex amplicon sequencing using Next Generation Sequencing (NGS)

**Library preparation, NGS and Bioinformatics data analysis.** The library preparation was done using Nextera XT DNA Library Preparation kit (Illumina Inc., USA) and quantification with Qubit™ dsDNA HS Assay Kit in Qubit 3.0 Fluorimeter (Invitrogen, USA). Final library was sequenced using MiSeq Reagent Kit v2 (Illumina Inc., USA). The raw Fastq reads were checked for quality using FastQC [33] and quality trimmed using Trimmomatic V 0.32 [34].

The filtered reads were mapped against reference sequence of F gene of NDV (JSD0812) using bowtie2 [35]. The consensus sequences were generated using SAMtools V 1.9 [36] and seqtk V 1.3 [37]. The workflow was performed for each sample to retrieve a consensus sequence for all samples. NDV F gene reference sequence was taken from the NCBI GenBank representing various genotypes of NDV (S1 Table in S1 File). A phylogenetic tree was constructed using a maximum likelihood in IQ-Tree using 1000 bootstrap similar to ones conducted by using W-IQ-TREE, which is a fast online phylogenetic tool for maximum likelihood analysis [38]. The tree was viewed using FigTree V 1.4.4 [39].

**Table 2. NDV Fusion (F) gene tiled fragments and designed PCR primers.**

|  | Primer | Fragment Position | Sequence (5' to 3') | Amplicon(bp) |
|---|---|---|---|---|
| Primer mix | NDVF_1561F | **Fragment 1 (95–546)** | TTGATGGCAGGCCTCTTGC | 497 |
|  | NDVF_IR2-2 |  | CATCTTCCCAACTGCCACTG |  |
|  | NDVF_IF2 | **Fragment 2 (469–1007)** | AGCATTGCTGCAACCAATGAGGC | 538 |
|  | NDVF_IR1 |  | GAGGTGTCAAGTTCTTCTATCAC |  |
|  | NDVF_IF1 | **Fragment 3 (882–1398)** | CCTAAATAATATGCGTGCCAC | 518 |
|  | NDVF_IR3 |  | CTCAGTTGAKATATCRAGATTGCCTG |  |
|  | NDVF_IF3-3 | **Fragment 4 (1287–1655)** | AGACGGGATAACTCTGAGGCT | 369 |
|  | NDVF_1561R |  | TTTGTAGTGGCTCTCATCTG |  |

## ND Outbreak (2021) investigation

There were multiple sporadic nationwide outbreaks of ND in 2021 in Nepal as reported by the Nepal Government Central Veterinary Laboratory (CVL) [40]. We conducted an outbreak investigation in Goldhunga, one of the highly affected areas near Kathmandu city. Our field team collected oral and cloacal swabs from dead chickens (n = 2) from an affected farm. We tested these samples for NDV and IAV and analyzed variant of the NDV.

## Developing thermostable I2- NDV vaccine (Ranigoldhunga™)

Based on the ND vaccine developed by the Australian Centre for International Agricultural Research (ACIAR, Australia) [41], we obtained the master seed (I2-NDV) from the University of Queensland and further developed the I2-NDV into a tablet formulation for greater efficacy and stability. We performed a thorough *in-vitro* (stability) and *in-vivo* (safety, clinical, and Genotype VII.2 challenge trial) tests to assess the efficacy of this new formulation (Ranigoldunga™). I2- NDV with Embryo Infectious Dose at 50% ($EID_{50}$) = $10^7$ or $10^8$ viral copies/ul per dose, was mixed with the stabilizers [hydrolyzed gelatin, skimmed milk and SPG (sucrose phosphate glutamate)], cryo-protectants and binders and lyophilized (freeze dried) [42]. We also prepared a freeze-dried formulation containing only I2- NDV suspended in skim milk (4%).

**Ranigoldunga™ vaccine stability assessment..**  Both formulations (lyophilized and tablet) of Ranigoldunga™ vaccine was subjected to stability tests in a stability chamber. Vaccines were tested at 4˚C, ambient temperature (20–25˚C), and 37˚C. $EID_{50}$ were measured at various time intervals (4, 7, 10, 14, and 28) days, and subsequently once a month from 2 to 6 months, to assess efficacy and stability of the vaccines. The $EID_{50}$ was measured in specific pathogen free (SPF) egg with the procedure described by WOAH guidelines [43].

**Ranigoldunga™ vaccine- in-vivo trial..**  In-vivo trials were conducted at the BIOVAC Nepal's animal testing facility located in Banepa (Nepal). Day old chicks (mix breed of *Gallus domesticus*) (n = 18) were selected and screened for 6 major poultry diseases [Newcastle disease virus (NDV), Influenza A Virus (IAV), Infectious bronchitis virus (IBV), Infectious Bursal disease (IBD), *Mycoplasma gallisepticum* (Mg), and *Mycoplasma synoviae* (Ms)] prior to the trial using PCR tests. Blood serum was also collected and screened for NDV antibodies using ELISA test. With all birds determined to be free from all six pathogens and showing no presence of NDV antibodies, they were grouped as described in Table 3.

Each vaccine dose contained around $10^{7-6.5}$ $EID_{50}$/ml (Table 3) and $10^8$ viral copies per ul (S2 Fig in S2 File). The viral copies were determined using a Real Time PCR test (S2 Fig in S2 File). Chickens were monitored daily for any visible clinical signs and symptoms. Serum was collected every week until week 7. An ELISA test was carried out every week to monitor antibody titers. Necropsy was carried out at the end of the trial to analyze post-vaccination anatomical effects on the bird.

**Ranigoldunga™ vaccine- field trial..**  Field trials were conducted at two sites (Goldhunga & Chaling) in the Kathmandu Valley. The Goldhunga farm was situated about 10 km south of the Kathmandu city. It has been in operation since 2018 and keeps around 3000 broiler

**Table 3. In-vivo (safety) trial- group (chickens), vaccine administration mode and dose with $EID_{50}$ values.**

| Group Number | Chicken ID | Vaccine Formulation Administered | Vaccine administration route | Vaccination dose | $EID_{50}$ Value |
|---|---|---|---|---|---|
| 1 | 1–6 | Negative Control | Ocular | 1 drop (40ul) | NA |
| 2 | 7–12 | Tablet Formulation | Ocular | 1 drop (40ul) | $10^7$ |
| 3 | 13–18 | Lyophilized Formulation | Ocular | 1 drop (40ul) | $10^{6.5}$ |

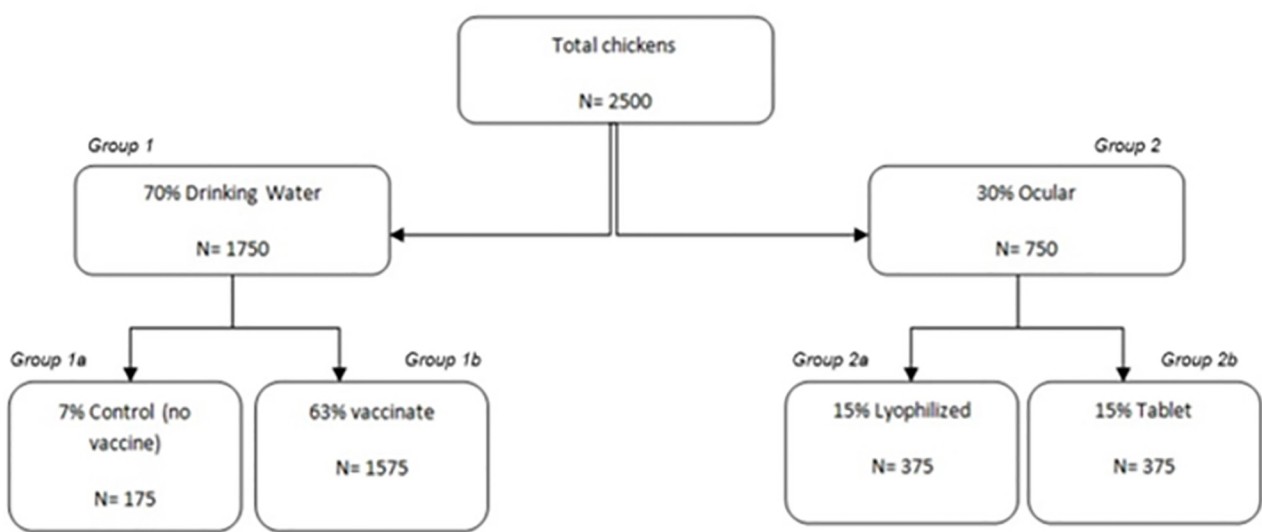

**Fig 3. Field trial of Ranigoldunga™ vaccine in two farms of the Kathmandu Valley.** Out of total 2500 chickens vaccination, 7% (n = 175) was selected as control group and 63% (n = 1575) as test group in drinking water and 15%, (n = 375) as test group for ocular administration and 15% (n = 375) test group for tablet formulation and assessed for the vaccine efficacy.

chickens (Cobb500) per production batch in its 2500 sq. ft. facility. The Chaling farm came into operation in 2019 and was located about 18 km north of the Kathmandu city. In its 3000 sq. ft. facility, it keeps around 3000 broiler birds (Cobb500). Prior to the vaccination trial, 1% of the chickens were randomly selected, sampled, and screened for NDV and IAV, including some water and feed samples. We tested our two different formulations (tablet and lyophilized) of the vaccine, which were administered through i) drinking water and ii) ocular application (S5 Table in S1 File).

Fig 3 shows segregation of birds during the clinical trials. Out of the 2500 birds in each farm (Goldhunga & Chaling), 70% (n = 1750, **Group 1**) were vaccinated with the lyophilized formulation. Of which 7% (n = 175, **Group 1a**) was selected as a control (no vaccine) group and 63% (n = 1575, **Group 1b**) were vaccinated through drinking water. The tablet vaccine was administered through intra-ocular method and was not tested in drinking water. To ensure maximum live virus vaccine exposure to the birds, we withheld access to drinking water for two hours prior to vaccine administration. Adequate vaccine dose in water was calculated based on average water consumption per bird [44].

**Group 2** (30%, n = 750) were vaccinated ocularly with the lyophilized formulation (15%, n = 375, **Group 2a**) and the tablet formulation (15%, n = 375, **Group 2b**) using a dropper [one drop (40ul) of vaccine- $EID_{50}$ ($10^{6.5}$ = Lyophilized, $10^7$ = Tablet)].

**Post vaccination assessment.** Birds were screened for the presence of live I2- NDV (one week after vaccination) and antibody titer response (within 3 weeks after vaccination). The chickens were segregated into different sections within the farm (Fig 4). A section was randomly selected and one bird per section was picked for sample collection (oral, blood) for I2-NDV PCR and antibody titer response assessment. To assess NDV antibody titer, hemagglutination inhibition (HI) titer were measured by using four hemagglutination (HA) units with a two-fold serial dilution as recommended by the WOAH (2012) [43]. To consider adequate coverage of the vaccine, we benchmarked attaining HI antibody titer of $\geq \log 2^3$ on more than 66% of the total vaccinated chickens. Total number of birds sampled from each section is shown in Table 4.

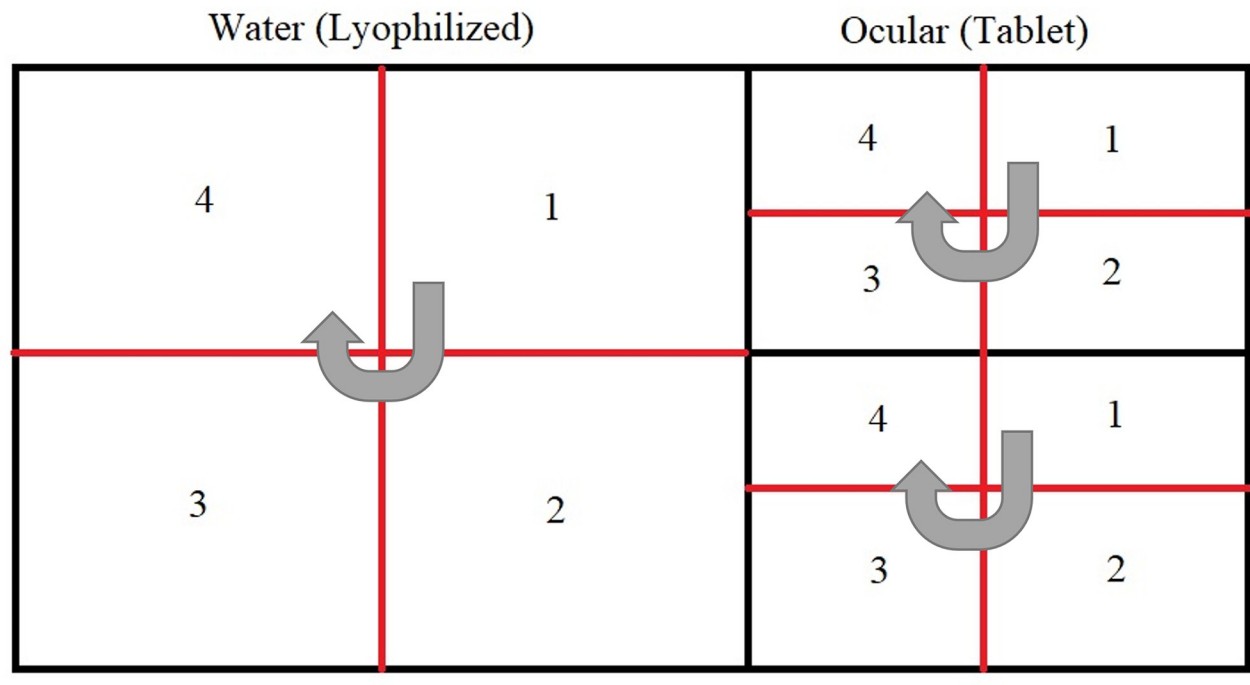

**Fig 4. Physical segregation of chickens in a trial farm-water section (lyophilized, Group 1a and 1b), ocular section (Tablet & Lyophilized, Group 2a and 2b) and additional quarters.** The direction of sampling starts with a selection of a random section followed by sections in the direction as indicated by the grey arrows.

**Ranigoldunga™ vaccine- challenge trial with Genotype VII.2..** The identified Genotype VII.2 NDV (2021 outbreak strain) was isolated, cultured in SPF egg, and harvested. The harvested fluid was tested for the lethal dose 50 ($LD_{50}$) value per the WOAH guidelines [45]. Birds were vaccinated with the Ranigoldunga™ vaccine on day 7. A mixture of 27 and 29 day-old chickens were segregated into non-vaccinated or control group (n = 13) and vaccinated or test group (n = 14). Both groups were given the outbreak NDV strain (Genotype VII.2) intramuscularly (0.2 ml of virus containing $10^{5.3}$ $EID_{50}$) at the BIOVAC's animal testing facility.

All chickens were screened for NDV using PCR in the first week of the trial. Daily body temperature and weight of the birds were recorded by a trained veterinarian. We classified

**Table 4. Number of birds sampled (with CI 95%) for the post vaccination assessment of live I2- NDV and antibody titer response at the two trial farms (Goldhunga and Chaling).**

| Farm | Group | Method of delivery | Total chickens |
|---|---|---|---|
| Goldhunga | Tablet formulation **Group 2b** | Ocular (combined) | 255 |
| | Lyophilized formulation **Group 2a** | | |
| | Lyophilized formulation **Group 1b** | Drinking water | 320 |
| | Unvaccinated **Group 1a** | Control population | 10 |
| Chaling | Tablet formulation **Group 2b** | Ocular | 191 |
| | Lyophilized formulation **Group 2b** | Ocular | 191 |
| | Lyophilized formulation **Group 1b** | Drinking water | 317 |
| | Unvaccinated **Group 1a** | Control population | 10 |

fever in birds if their body temperature was above normal (105.8 ˚F to 106.7 ˚ F) [46]. We recorded clinical manifestations and mortality of the trial birds for 42 days.

## Results

### Biosecurity and biosafety assessment of commercial and backyard farms

**Commercial farms.** The top three chicken breeds found in the commercial farms were Cobb 500 (45%), Hyline brown (32.5%) and Lohmann brown (17.5%). 97.5% of the farms used at least one type of disinfectant to clean their farms, and 67.5% used at least one form of personal protective equipment (PPE) while working inside their farm. All surveyed farms vaccinated their chickens against ND, except two farms [Kailali district (n = 1) and Nawalparasi district (n = 1)] (Table 5).

**Backyard farms.** Giriraj (27.5%), Kuroiler (22.5%) and Sakhini and Pwakh Ulte (12.5% each) were the most common poultry breeds found in backyard farms. Only small numbers of farms used disinfectant (< 32.5%) and PPE (<15%) (Table 6). Only 2.5% of the farms vaccinated their chickens against ND. None of the backyard farms in Jhapa district consented to participate in the study and could not be included, bringing the total number of backyard farms sampled to 36.

We observed limited biosafety and biosecurity practices in the commercial farms (Table 7). Although all 40 commercial farms used disinfectant to clean and the majority of farms (n = 39) also followed proper hand washing practices after working with stock, only 32.5% of the farms (n = 13) used some kind of PPE when handling waste. Four farms (10%) used comprehensive biosafety measures in their farms. All 40 farms vaccinated their stock against at least one of the diseases. Ten farms reported their poultry flock interacted with wild birds, five farms obtained poultry from multiple sources, and only three stored multiple species of poultry in one enclosure.

Most of the farmers (43%) thought IA (bird flu) was the main poultry disease, and only few (3%) thought ND caused disease in their birds. Most of the farmers (51%) were not aware of any other poultry disease (Fig 5).

**Serological screening for antibody against NDV and IAV.** Of the 600 commercial chickens tested from 40 farms in 10 districts of Nepal for the presence of NDV and IAV antibodies in their serum, NDV antibody was detected in the majority of the samples (n = 381, 64%, 28 farms). Comparatively, IAV antibody was detected in fewer number of samples (n = 125, 21%, 11 farms). Similarly, in 108 backyard chicken tested from 36 farms of 9 districts of Nepal, NDV antibody was detected in 35% (n = 38, 7 farms), and IAV antibody was present in 16% (n = 17, 3 farms) of the birds (Table 5).

### Molecular screening for presence of NDV and IAV

Pooled oral and cloacal samples were screened for NDV and IAV using the in-house designed NDV-ISO and IAV-ISO primers respectively. Out of the 40 commercial farms, 31(78%) and 15(38%) were PCR positive for NDV and IAV respectively. Similarly, out of the 36 backyard farms, 6 (17%) and 1 (3%) were PCR positive for NDV and IAV respectively.

### Detected NDV genotypes (strain)

Samples with detectable NDV were selected and 4 fragments of F-gene were further amplified using PCR and sequenced. Not all 4 fragments were successfully amplified. Only fragment 2 was consistently amplified in most of the samples from commercial (11/31, 33%) and backyard

Newcastle disease burden in Nepal and vaccine efficacy

**Table 5. Biosecurity practices in commercial farms in ten districts of Nepal along with antibody ELISA results for NDV and IAV.** ELISA results provide an insight into any possible correlations between biosecurity measures and presence of diseases.

| Districts (4 farms per district) | Chicken Breed | | | | | Biosecurity Measures | | | | | | Antibody ELISA | | | |
|---|---|---|---|---|---|---|---|---|---|---|---|---|---|---|---|
| | Broilers | | Layers | | Other | At least one Disinfectant Used | | At least one PPE Used | | ND Vaccines | | ELISA_NDV | | ELISA_IAV | |
| | Cobb 500 | H&N Brown Nick | Hyline Brown | Lohmann Brown | Giriraj | Yes | No | Yes | No | Used | Not Used | Positive | Negative | Positive | Negative |
| Bhaktapur | 2 | 0 | 1 | 1 | 0 | 4 | 0 | 4 | 0 | 4 | 0 | 1 | 3 | 3 | 1 |
| Chitwan | 2 | 0 | 1 | 1 | 0 | 4 | 0 | 3 | 1 | 4 | 0 | 3 | 1 | 2 | 2 |
| Dang | 1 | 1 | 1 | 1 | 0 | 4 | 0 | 2 | 2 | 4 | 0 | 4 | 0 | 0 | 4 |
| Jhapa | 0 | 0 | 4 | 0 | 0 | 4 | 0 | 3 | 1 | 4 | 0 | 4 | 0 | 0 | 4 |
| Kailali | 1 | 0 | 1 | 1 | 1 | 3 | 1 | 2 | 2 | 3 | 1 | 2 | 2 | 1 | 3 |
| Kathmandu | 3 | 0 | 1 | 0 | 0 | 4 | 0 | 2 | 2 | 4 | 0 | 3 | 1 | 3 | 1 |
| Lalitpur | 3 | 0 | 0 | 1 | 0 | 4 | 0 | 2 | 2 | 4 | 0 | 0 | 4 | 0 | 4 |
| Morang | 2 | 0 | 2 | 0 | 0 | 4 | 0 | 3 | 1 | 4 | 0 | 3 | 1 | 1 | 3 |
| Nawalparasi | 2 | 0 | 1 | 1 | 0 | 4 | 0 | 2 | 2 | 3 | 1 | 4 | 0 | 1 | 3 |
| Sunsari | 2 | 0 | 1 | 1 | 0 | 4 | 0 | 2 | 2 | 4 | 0 | 4 | 0 | 0 | 4 |
| *Percentage* | *45.0* | *2.5* | *32.5* | *17.5* | *2.5* | *97.5* | *2.5* | *67.5* | *35.0* | *95.0* | *5.0* | *70.0* | *30.0* | *27.5* | *72.5* |

https://doi.org/10.1371/journal.pone.0280688.t005

**Table 6. Biosecurity practices in backyard farms(n = 36) of Nepal along with antibody ELISA results for NDV and IAV.** ELISA results provide an insight into any possible correlations between biosecurity measures and presence of diseases.

| Districts | Chicken Breed | | | | | | | Biosecurity Measures | | | | | | Antibody ELISA | | | |
|---|---|---|---|---|---|---|---|---|---|---|---|---|---|---|---|---|---|
| (4 farms per district) | *Broilers* | *Breeds* | | | | | | *At least one Disinfectant Used* | | *At least one PPE Used* | | *Vaccines Used* | | *ELISA_NDV* | | *ELISA_IAV* | |
| | *Cobb 500* | *Giriraj* | *Kadaknath* | *Kuroiler* | *Sakhini* | *Pwakh Ulte* | | *Yes* | *No* | *Yes* | *No* | *Yes* | *No* | *Positive* | *Negative* | *Positive* | *Negative* |
| Bhaktapur | 1 | 2 | 0 | 1 | 0 | 0 | | 0 | 4 | 0 | 4 | 0 | 4 | 1 | 3 | 0 | 4 |
| Chitwan | 0 | 1 | 0 | 1 | 1 | 1 | | 1 | 3 | 1 | 3 | 0 | 4 | 0 | 4 | 0 | 4 |
| Dang | 0 | 3 | 1 | 0 | 0 | 0 | | 2 | 2 | 1 | 3 | 0 | 4 | 2 | 2 | 0 | 4 |
| Jhapa* | NA | NA | NA | NA | NA | NA | | NA | NA | NA | NA | NA | NA | NA | NA | NA | NA |
| Kailali | 1 | 0 | 1 | 0 | 1 | 1 | | 0 | 4 | 0 | 4 | 0 | 4 | 3 | 1 | 1 | 3 |
| Kathmandu | 1 | 3 | 0 | 0 | 0 | 0 | | 2 | 2 | 0 | 4 | 0 | 4 | 0 | 4 | 0 | 4 |
| Lalitpur | 0 | 0 | 0 | 2 | 1 | 1 | | 2 | 2 | 0 | 4 | 0 | 4 | 0 | 4 | 0 | 4 |
| Morang | 0 | 2 | 1 | 0 | 0 | 1 | | 1 | 3 | 2 | 2 | 0 | 4 | 0 | 4 | 0 | 4 |
| Nawalparasi | 0 | 0 | 0 | 3 | 1 | 0 | | 2 | 2 | 1 | 3 | 1 | 3 | 0 | 4 | 1 | 3 |
| Sunsari | 0 | 0 | 0 | 2 | 1 | 1 | | 3 | 1 | 1 | 3 | 0 | 4 | 1 | 3 | 1 | 3 |
| *Percentage* | *7.5* | *27.5* | *7.5* | *22.5* | *12.5* | *12.5* | | *32.5* | *57.5* | *15* | *75* | *2.5* | *87.5* | *17.5* | *72.5* | *7.5* | *82.5* |

*Farms in Jhapa did not consent to sampling and are thus listed as NA. Total farms questioned are n = 36.

(4/6, 67%) farms (S1 Fig in S2 File). These amplified fragments were extracted and cleaned for DNA sequencing in the next generation sequencing platform (MiSeq, Illumina, USA).

We constructed a phylogenetic tree (Fig 6) based on F-gene sequence from samples with detectable NDV (commercial farms = 5, backyard farms = 2) along with archived sequence data from the NCBI GenBank of all known NDV genotypes. Maximum likelihood phylogeny tree showed that the all-commercial farm (n = 5) NDV F-gene sequences clustered with non-virulent Genotype II. This genotype also consisted of vaccine strains- Lasota, B1 and F strain. Four of these commercial samples [BCCHT2_2018 (NCBI accession # MZ087886), NWLP4_2018 (NCBI accession # MZ087887), LTP3_2018 (NCBI accession # MZ087888) and BCDNG1_2018 (NCBI accession # MZ087890)] clustered together with the KU159667 (USA)

**Table 7. Biosafety (left) and Biosecurity (right) results of commercial farms (n = 40) sampled by BIOVAC.**

| Biosafety *Results* | | Biosecurity *Results* | |
|---|---|---|---|
| *Chosen variables* | *N (%)* | *Chosen variables* | *N (%)* |
| Disinfectant Used | 40 (100) | Veterinary care in the last one year | 40 (100) |
| Hand washing practices after working with stock | 39 (97.5) | Private veterinary care | 40 (100) |
| PPE Used | 25 (67.5) | Vaccines available on demand | 40 (100) |
| Used only one type of biosafety measures (hand washing, showering, footbaths, PPE for visitors, fencing, etc.) | 23 (57.5) | Important to report suspicious deaths | 37 (92.5) |
| PPE used when handling waste | 13 (32.5) | Vet visit frequency as needed | 34 (85) |
| Knowledge among workers regarding zoonotic potential of some poultry diseases | 11 (27.5) | Quarantine facility for new birds | 33 (82.5) |
| Used four types of biosafety measures (hand washing, showering, footbaths, PPE for visitors, fencing etc.) | 4 (10) | Enclosure cleaned as needed | 31 (77.5) |
| | | Stock interacted with wild birds | 10 (25) |
| | | Stock obtained from multiple sources | 5 (12.5) |
| | | Multiple species stored in one enclosure | 3 (7.5) |
| | | Farm inspection by government officials in the past year | 2 (5) |
| | | Refrigerator present on site for vaccine storage | 2 (5) |

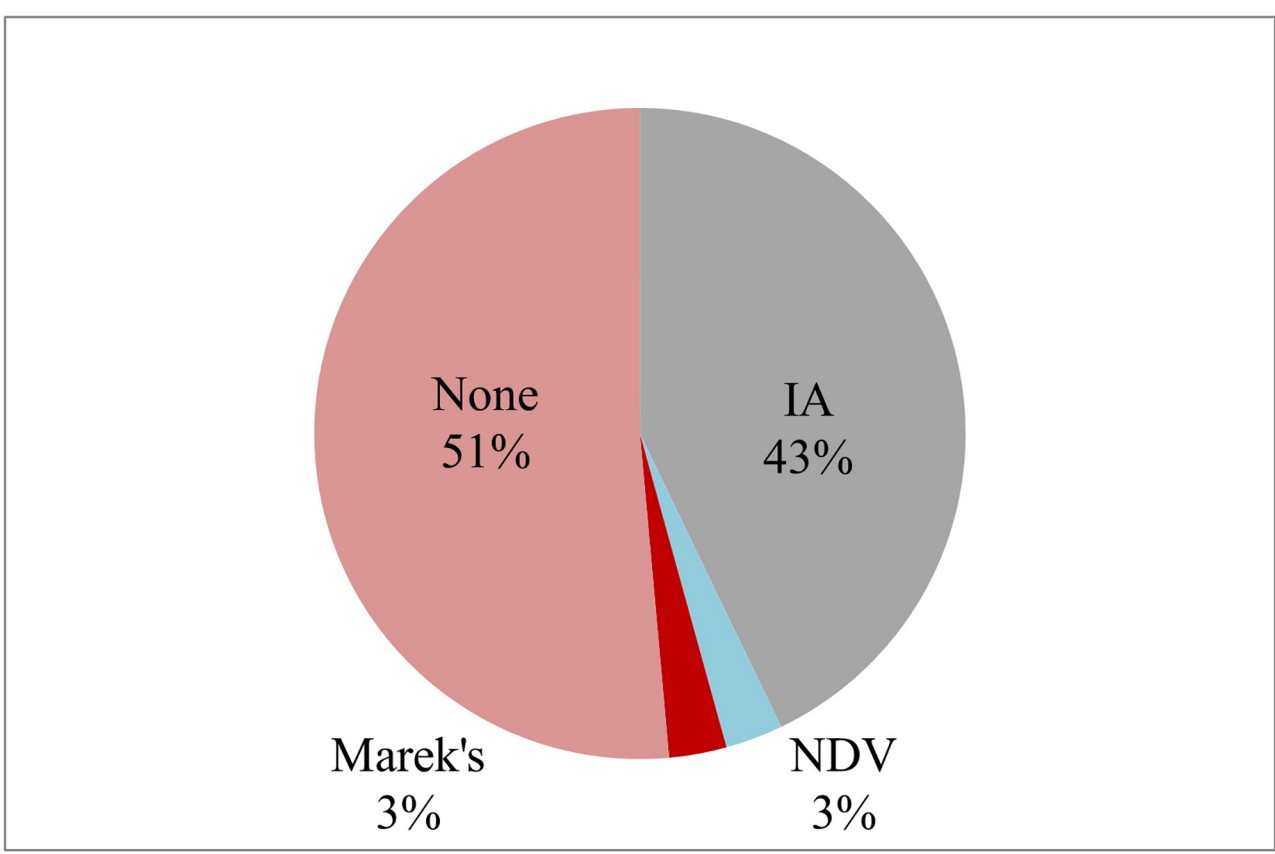

**Fig 5. Commercial poultry farmers' perception to diseases that were responsible for outbreaks in their poultry farms.** Majority (43%) thought Bird flu was responsible for disease outbreaks, only few (3%) blamed ND for making their birds sick.

and the MH996947 (Nigeria) NCBI references. The remaining sample [SNS4_2018 (NCBI accession # MZ087889)] was closely related to the JN942019 (Peru) NCBI reference. Meanwhile NDV from the backyard farms (n = 2) samples clustered with the MH996911 (Nigeria) NCBI reference in the Genotype I group. The outbreak strain, VCS1_F (MZ087884), from the Kathmandu Valley belonged to the Genotype VII clade.

### Newcastle disease outbreak investigation (2021)

Our investigation in one of the diseased farms in Goldhunga (Kathmandu valley) in 2021 found NDV to be the cause of the outbreak. F-gene sequence analysis characterized the detected NDV as a Genotype VII.2 variant (NCBI accession # MZ087884.1) (Fig 7).

### Ranigoldunga™ vaccine stability assessment

Our preliminary data showed that both the formulations (tablet & lyophilized) of the Ranigoldunga™ vaccine were stable for 30 days at ambient temperature (10–22˚C), 8 days at 37˚C, and an estimated 1 year at 4˚C (Fig 8) with $EID_{50} > 10^6$.

### Ranigoldunga™ vaccine- in-vivo trial

The antibody titer after the **Ranigoldunga™** vaccination peaked (titer peaking at 2000–4000) at week 4 and gradually subsided in subsequent weeks (Fig 9). The data showed the need for a booster dose after week 7 to revamp NDV antibody titers.

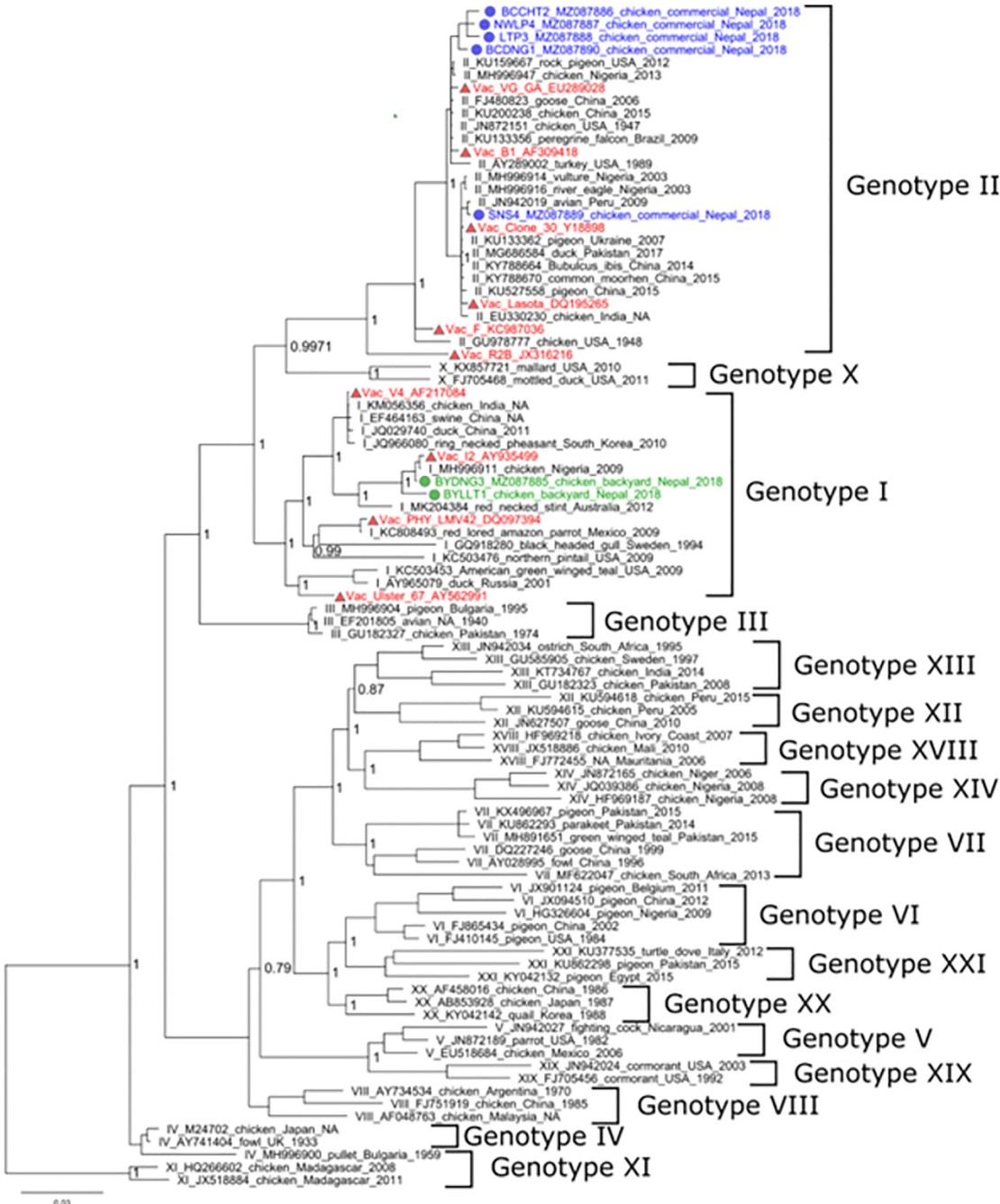

**Fig 6. Phylogenetic analysis of the nucleotide sequences from the amplified products of NDV fusion genes (521 bp) for commercial and backyard chickens.** A phylogenetic tree was constructed using a maximum likelihood in the IQTree with 1000 bootstrap replicates. The tree was viewed using Fig Tree V 1.4.4. NDV strains are represented in colors based on their detection source- commercial chickens (Blue), backyard chickens (green), vaccine strains (red) and the outbreak strain (brown).

## Vaccine efficacy- field trial

We found the **Ranigoldunga™** vaccine to be very efficacious, with one of the farms (Gold-hunga) showing over 88% efficacy (HI titer >Log $2^3$) both when administered ocularly or used in drinking water. In the second trial farm (Chaling), although the efficacy was observed lower

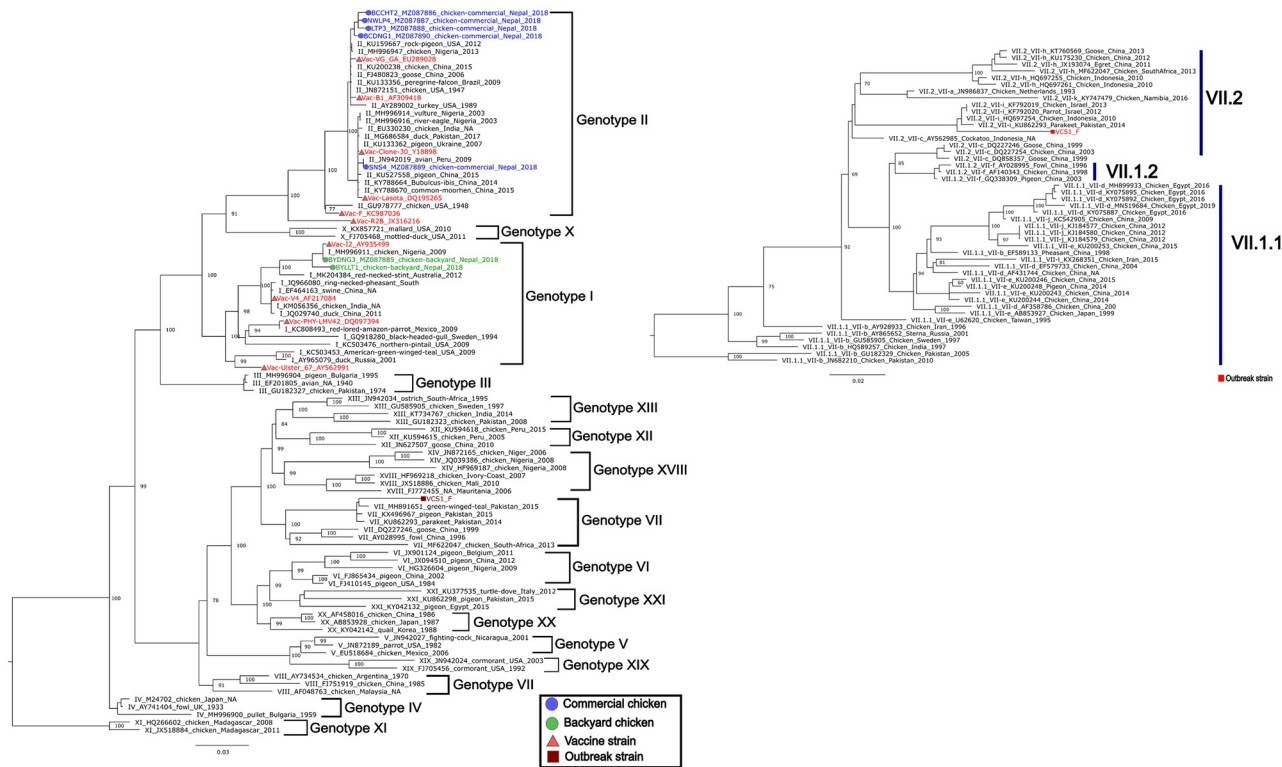

**Fig 7. Phylogenetic analysis of the partial NDV Fusion gene (744 bp) of the 2021 outbreak strain (brown color, MZ087884.1) clustered it in the Genotype VII group.** Further phylogenetic analysis placed the outbreak strain (red) in the Genotype VII.2 group [47].

(69% for ocular; 79% for drinking water) compared to Goldhunga farm, it was still considered efficacious based on the WOAH guideline of attaining at least 66% or higher (HI titer >Log $2^3$) (S7 Table in S1 File) [48].

We also recorded morbidity and mortality at these two farms (S8 Table in S1 File). Of all the vaccinated chicken (n = 2500) at the Goldhunga farm, 500 chickens (19%) died in a course of 9 weeks. During the first few weeks, salmonella infection was the major cause of mortality. In the subsequent weeks, chickens started wheezing, showed signs of labored breathing, lethargy and loss of appetite, symptoms associated with chronic respiratory disease (CRD). Swab collected from the post-vaccination chickens resulted in <2% of chickens developing signs of ND. Interestingly, a ND outbreak was ongoing in Kathmandu at the time of this trial [40]. Out of 2500 chickens at the Chaling farm, 197 chickens (8%) chickens died over the period of six weeks. Overcrowding and trampling caused early mortality as a sudden drop in temperature resulted in chickens brooding in tight spaces. Ascites and CRD were other main cause of morbidity and mortality in weeks leading to end of production cycle (45 days). Morbidity and mortality of both farms are shown in S8 & S9 Tables in S1 File.

### Vaccine efficacy- challenge trial with genotype VII.2 strain

In the controlled group (non-vaccinated, challenged), we recorded fever (108°F) in birds after two days of challenge dose administration. In the test group (vaccinated, challenged), birds' average body temperature remained within the normal range (106°F) (Fig 10). Other clinical symptoms atypical of ND were also observed in the controlled group.

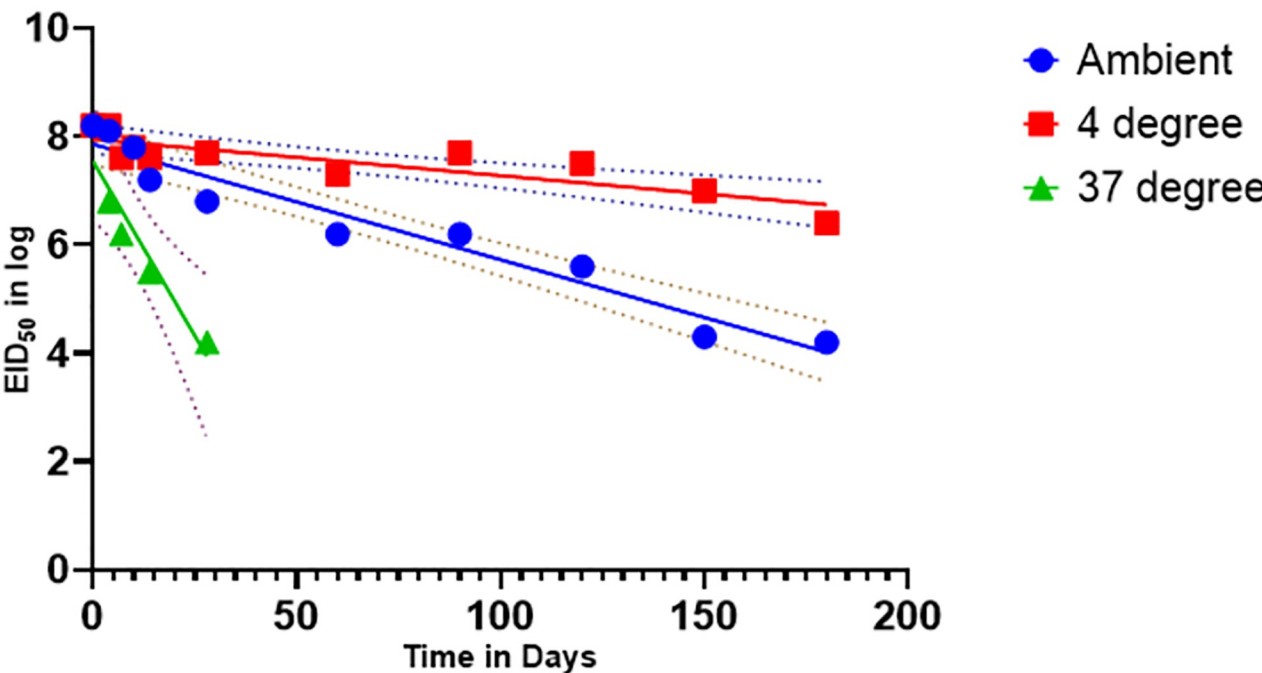

**Fig 8. Ranigoldhunga™ vaccine stability tested at different temperature within a range of relative humidity (40–60%).** Ambient temperature (blue) ranging from 20–25°C, 4 ˚C (red) and 37 ˚C (green) were three different temperatures the vaccine was tested for. Recorded stability at $EID_{50}$ is represented by the solid lines, and plotted stability (CL- 95%) derived from a mean value calculation is shown with the dotted lines.

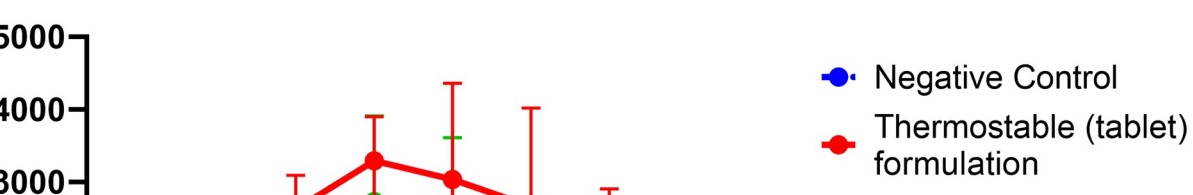

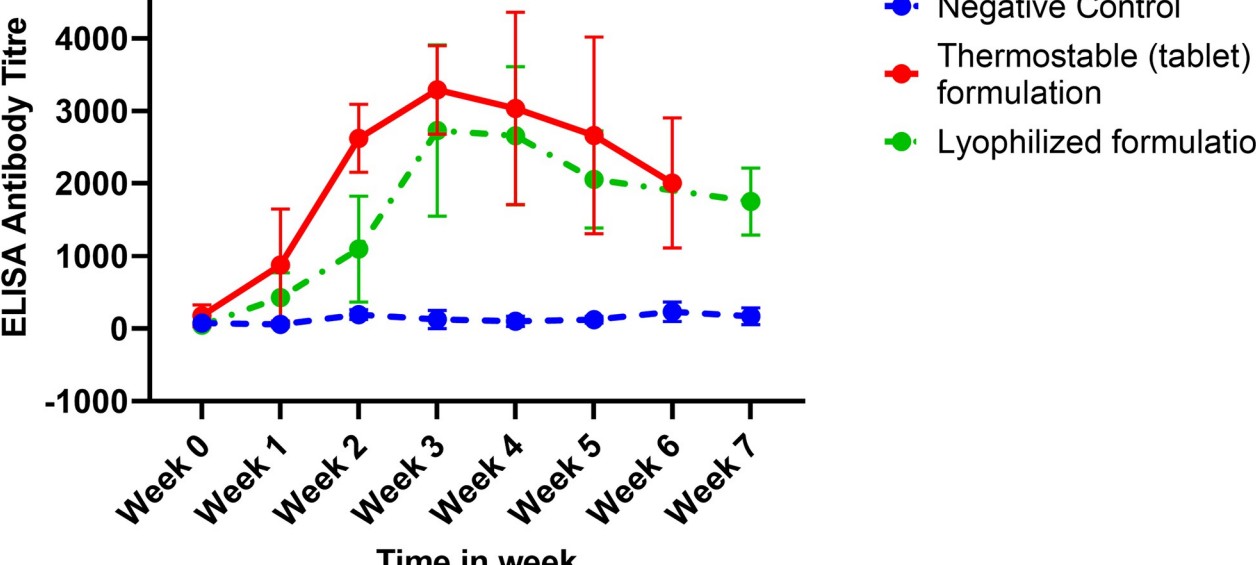

**Fig 9. In-vivo measurement of NDV antibody titer after vaccination with the Ranigoldunga™.** The Tablet formulation elevated NDV antibody titer to 4000 based on ELISA assessment. The graphs highlight antibody titer in chickens vaccinated with thermostable (tablet) formulation, lyophilized formulation, and negative control (unvaccinated chickens) from week 0 to week 7.

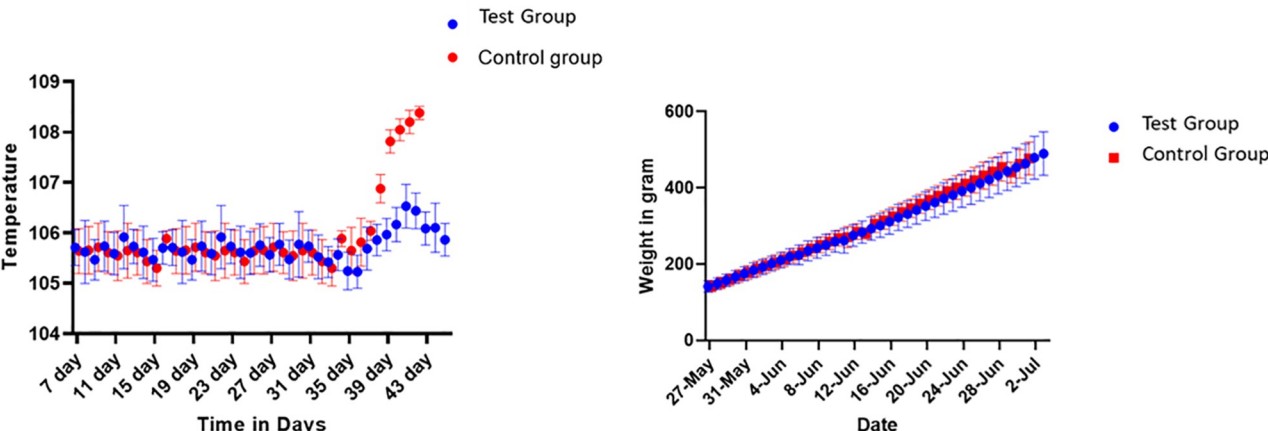

**Fig 10. Ranigoldhunga™ vaccine challenged trial in the test group (n = 14) and the Control group (n = 13).** Measurement of increase in temperature (left) and change in body weight (right) of the chickens that were challenged intramuscularly (0.2 ml) with a total dose of Ge Genotype VII.2 105.3 $EID_{50}$.

**Table 8. Summary of challenge trial conducted to test vaccine efficary agaisnt Genotype VII.2.**

| Group | Number of chickens | Method of challenge | Mortality | survival percentage % |
|---|---|---|---|---|
| Control (Challenged, Non-vaccinated) | 13 | intramuscular | 13/13 | 0 |
| Test (Challenged, Vaccinated) | 14 | intramuscular | 0/14 | 100 |

Post-vaccination assessment data is shown in S3 Table in S1 File.

We did not observe any severe symptoms of ND (neck and leg paralysis) in birds that required euthanasia. However, all the birds in the controlled group died within 3–5 days of receiving the challenge dose (Table 8). The necropsy of the birds showed visible signs of blood clots in the trachea, a swelling of the intestines, and nervous system damage. **Ranigoldunga<sup>TM</sup>** vaccine was highly effective against Genotype VII.2 of NDV.

## Discussion

RNA viruses, like NDV, are highly mutable and pose a great threat due to their changing virulence [49]. The 2021 outbreak of ND in Nepal was particularly devastating with thousands of birds lost across Nepal. While commercial poultry farmers are aware of the need to vaccinate against ND, most of the backyard farmers either do not know or do not have access to the ND vaccine [50–52].

Outbreaks of ND and IA frequently occur in low resource countries like Nepal [53]. However, prior to this study, the disease burden and epidemiological dynamics in Nepal were never studied. We have conducted a nationwide ND and IA prevalence study by collecting samples from representative commercial and backyard poultry farms from across the major poultry production hubs of Nepal [29] using both serological and molecular assessments. Seroprevalence of NDV and IAV in commercial farms were 70% (n = 28) and 27.5% (n = 11) respectively, and 17.5% (NDV) and 7.5% (IAV) respectively in backyard farms. Out of the 40 commercial farms, we were able to detect live NDV in 31 farms (78%) and IAV in 15 farms (38%) (Fig 11) by using molecular techniques. Similarly, out of 36 backyard farms, NDV and IAV were detected in 3 each (8%) (Fig 12) Usage of live virus vaccines for ND and the

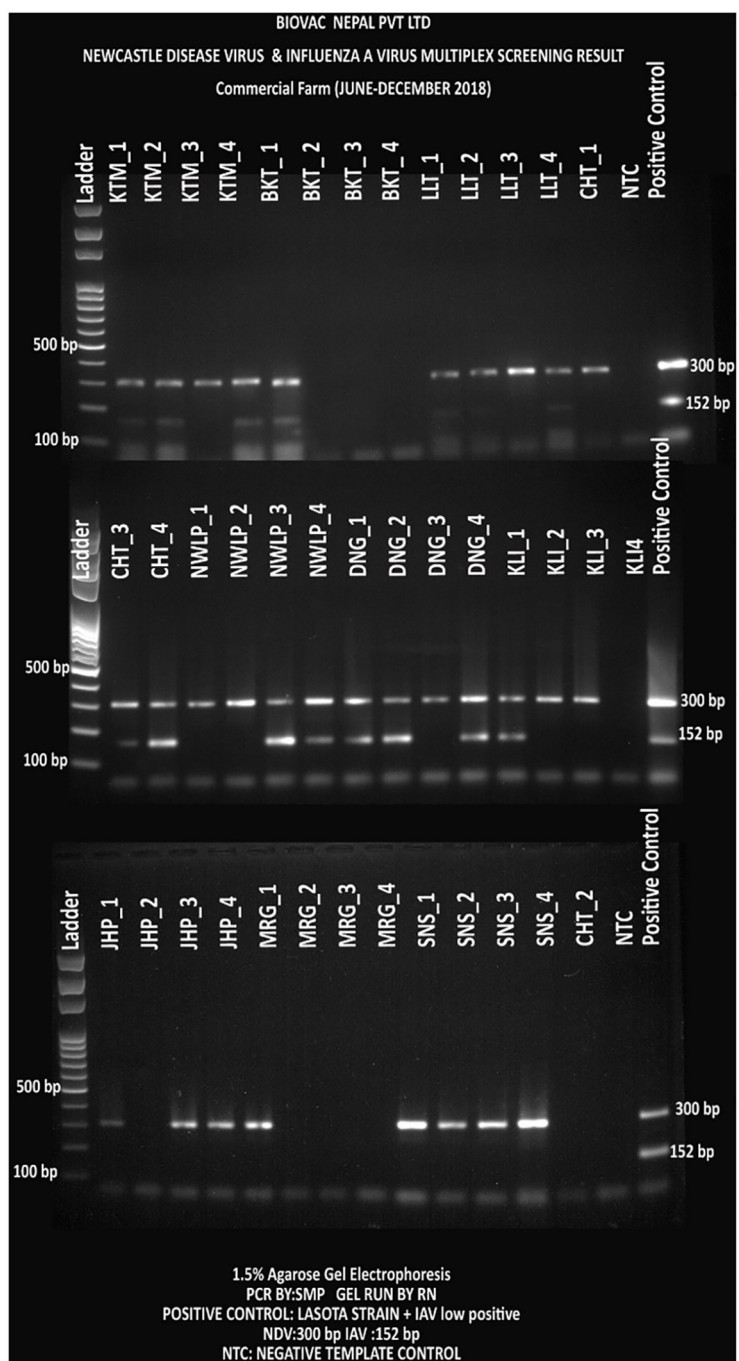

**Fig 11. NDV and IAV screening PCR test in the commercial farms (n = 40)–a multiplex PCR test, NDV was detected with 300 bp and IAV with 152 bp amplicons.** Each district had three letter code and included four farms (numbered 1 to 4). Each sample represents pooled samples from a single farm. The gel was run with ladder in the first well; and negative and positive controls in the last wells (from left to right). (KTM = Kathmandu, LLT = Lalitpur, CHT = Chitwan, NWLP = Nawalparasi, DNG = Dang, KLI = Kailali, MRG = Morang, and SNS = Sunsari Districts).

detection of Genotype II NDV in commercial farms resulted in a high seroprevalence of NDV. However, the government of Nepal has banned usage of unverified and unlicensed vaccines against IA but still used by poultry farmers, and a 21% seroprevalence of IA probably means either there is a high, under-reported IAV infection and/or illegal use of IAV vaccines.

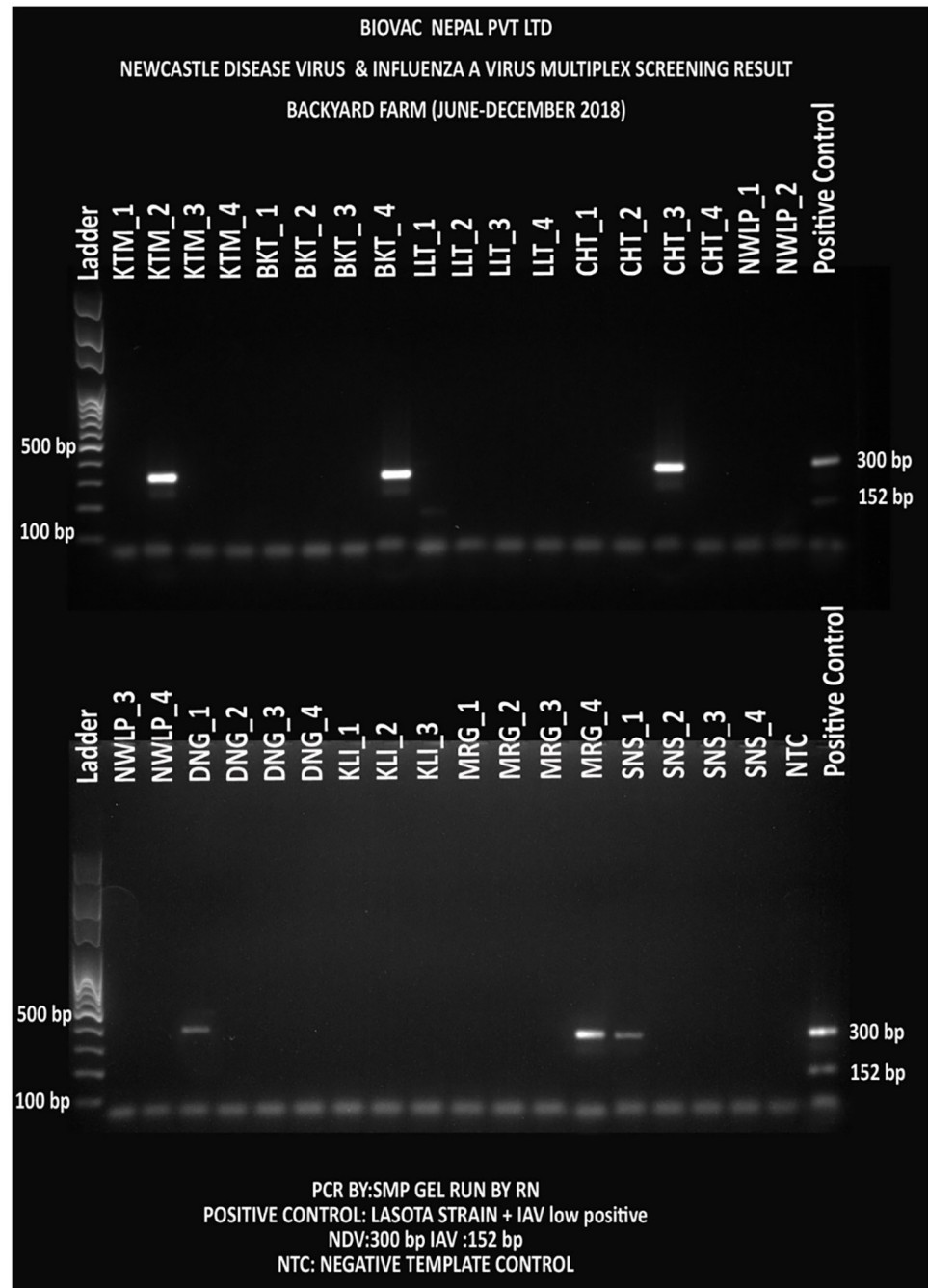

**Fig 12. NDV and IAV screening PCR test in the backyard farms (n = 36)–NDV and IAV were detected with 300 bp and 152 bp PCR amplicons respectively.** Each district were assigned with a three-letter code and had four farms (numbered 1 to 4). Each sample represents a farm pooled in a single tube and screened. The gel was run with ladder in the first well and negative and positive controls in the last wells (from left to right). KTM = Kathmandu, LLT = Lalitpur, CHT = Chitwan, NWLP = Nawalparasi, DNG = Dang, KLI = Kailali, MRG = Morang, and SNS = Sunsari Districts.

In backyard farms, we detected the Genotype I NDV strain, which has not been previously reported in Nepal, indicating potential identification of native, endemic strain of the virus. The NDV strains, if lentogenic, can be used as vaccine targets. Our 2021 ND outbreak investigation identified Genotype VII.2 as the causative strain (variant).

Understanding the disease burden, gathering information on viral strains, and administering effective vaccines can help mitigate and prevent frequently occurring ND outbreaks in poultry production in low resource settings [53]. Our survey indicated that farmers were not aware of ND (as much as IA) and hence were not using vaccines against ND. This finding was especially true for the backyard poultry farmers (Fig 5).

One of the most effective disease preventive practices in poultry production is to have adequate biosafety and biosecurity measures. Biosafety refers to the containment principles, technologies, and practices that are implemented to prevent unintentional exposure to pathogens and toxins, or their accidental release. Biosecurity refers to the institutional and personal security measures designed to prevent the loss, theft, misuse, diversion or intentional release of pathogens and toxins [54]. Although immunization in the commercial farms were high (98%), biosecurity measures were mostly inadequate, with only two-thirds (65%) of the farms using some form of PPE and only 10% having comprehensive biosafety protection (Tables 5 & 7). It was evident that poultry health was paramount to the commercial farms because of the high vaccination rates and veterinary care, but self-monitoring or other biosafety measures were not prioritized. Overall, poor biosafety and biosecurity practices were observed in the commercial farms (Table 7). All 40 farms vaccinated their stock against at least one of the poultry diseases. Ten farms reported their poultry flock interacted with wild birds which are known to carry various poultry diseases like IA, ND, and Infectious Bursal Disease [55–57]. This finding was also validated by the fact that a mere 27.5% of the farms reported having knowledge of zoonotic disease threats caused by poultry pathogens. Having little or no knowledge of impacts of zoonoses may explain the lapse in biosafety and biosecurity measures. Although all 40 farms used disinfectants to clean, and the majority of farms (n = 39) followed proper hand washing practices after working with stock, only a few of the farms (n = 13, 32.5%) used some kind of PPE when handling waste. Only 4 farms (10%) used a comprehensive biosafety measure in their farms. Five farms obtained chicken (day old chicks) from multiple sources, and only three stored multiple species of poultry in one enclosure. (Table 7).

Additionally, biosafety and biosecurity measures were very poorly practiced and implemented in the backyard farms due to lack of knowledge and experience in rearing poultry [58]. Less than a third of the farms (32.5%) used at least one type of disinfectant to clean chicken housings. Only 15% reported to ever vaccinate their stock, and only one farm out of the 36 (2.5%) reported using vaccines. Backyard poultry are more resilient to diseases [59], but their close proximity to commercial farms (sometimes within less than 100m of each other—as discovered during our field work) can lead to commercial poultry being affected by disease outbreaks that originate in backyard farms [44]. And lack of biosafety practices can lead to zoonotic spillover of possible highly pathogenic avian influenza (HPAI) into backyard poultry farmers, jeopardizing their health.

Animal health services, especially laboratory-based diagnostics, are not well developed in Nepal [60]. Poultry farms do not have reliable laboratories to have outbreaks investigated in a timely fashion. Although immune-chromatography (rapid)-based diagnostic kits are readily available, they are often not reliable [61]. Molecular-based detection methods (like PCR) are more accurate and sensitive to detect pathogens, such as NDV and IAV [62]. Clinical symptoms caused by NDV and IAV are very similar and hence extremely difficult to differentiate by clinical symptoms alone. IAV, especially HPAI, often get lot of attention from public health officials with active Avian Influenza surveillance programs implemented

throughout the country. With a challenging differential diagnosis between IAV and NDV, the implication to the farmer is tremendous—if erroneously diagnosed, as highly infectious strains of IAV like HPAIs lead to culling of all poultry within 5 km radius of an infected farm. Having relatively cheap NDV and IAV molecular diagnostic and characterization tools is going to help address this challenge. With many PCR labs now set up during SARS-CoV-2 pandemic period [63], such tests can easily be run (with proper adaptation) in molecular labs located across the country, increasing greater access to diagnosis for early and effective interventions.

Thermostable I2-ND vaccine (Ranigoldunga™) in tablet formulation is highly effective against NDV, including a virulent strain of Genotype VII.2 that devastated the majority of Kathmandu poultry farms in 2021 [63]. In-vivo trials with both the Ranigoldunga™ formulations (tablet and lyophilized) performed very well with a relatively high protective titer response (>3000) against NDV (Fig 8). A single dose of Ranigoldunga™ at day 7 will be enough to protect short cycle broiler breed such as Cobb 500 that have turnovers in 45 days. Currently, general practice is to give three different doses of ND vaccine(s) at day 7, 18, and 22 [6], considerably increasing the cost of immunization. Differences in vaccine efficacies observed between the two field trial sites could be due to inefficiencies in vaccine delivery (such as eye dropper inaccuracies) and/or immune-compromised chickens with morbidity caused by other infections (S9 Table in S1 File).

The live ND vaccines are also mostly of lentogenic and mesogenic strains that require cold chain transportation, and currently only one I-2 strain based vaccine produced by the National Vaccine Production Lab (NVPL) is used for the vaccinating backyard poultry [6]. We believe vaccines like Ranigoldunga™ can be used for commercial and backyard poultry and increase the accessibility of immunization against NDV in places where maintaining cold chain for transportation and storage is a challenge. This would reduce the economic burden on poultry farmers and help to significantly boost production in one of the major, self-sustaining industry of Nepal.

## Supporting information

**S1 File.**
(DOCX)

**S2 File.**
(DOCX)

**S3 File.**
(PDF)

## Acknowledgments

We would like to thank our collaborators from the University of Queensland (Australia), Pirbright Institute (UK) and Universidad de Castilla La Muncha (Spain) for their valuable technical support. We would also like to thank previous and current Ambassadors of Australia to Nepal (HE Glen White, Peter Budd, Felicity Volk) for facilitating technical support and encouraging us. This work would have been impossible without ACIAR providing us with NDV I2 master seed- we express our gratitude. Special thanks goes to Professor Joanne Meers of the University of Queensland for helping us with the vaccine seed, and providing us with valuable technical assistance. Various components of our effort were supported by PSI (the Netherlands) and InnovationXchange grant from DFAT (Australia), we would like to acknowledge and thank them. Thank you, Dr David Bunn of the University of

California-Davis, for giving us the idea of setting up the Animal vaccine facility (BIOVAC Nepal Pvt. Ltd.) in Nepal. We would like to thank Dr. Rupendra Chaulagain, Dr. Prakash Adhikari, Dr. Sonu Adhikari, Arjun Bhujel, Deepesh Oli, Samita Raut, and Dhiraj Puri for their invaluable contribution in the field work and sample collection. We would like to thank everyone at the Center for Molecular Dynamics Nepal and Intrepid Nepal for all their help. And finally, we would like to show our appreciation to the local and central government agencies of Nepal for all their assistance and encouragement.

## Author Contributions

**Conceptualization:** Rajindra Napit, Prajwol Manandhar, Ajay N. Sharma, Dibesh B. Karmacharya.

**Data curation:** Rajindra Napit, Ajit Poudel, Saman M. Pradhan, Sajani Ghaju, Ajay N. Sharma, Jyotsna Joshi, Suprim Tha, Kavya Dhital.

**Formal analysis:** Rajindra Napit, Ajit Poudel, Saman M. Pradhan, Sajani Ghaju, Suprim Tha, Rajesh M. Rajbhandari.

**Funding acquisition:** Dibesh B. Karmacharya.

**Investigation:** Rajindra Napit, Ajit Poudel, Saman M. Pradhan, Prajwol Manandhar, Ajay N. Sharma, Jyotsna Joshi, Suprim Tha, Kavya Dhital, Udaya Rajbhandari, Amit Basnet.

**Methodology:** Rajindra Napit, Ajit Poudel, Saman M. Pradhan, Prajwol Manandhar, Sajani Ghaju, Ajay N. Sharma, Jyotsna Joshi, Suprim Tha, Kavya Dhital, Udaya Rajbhandari, Amit Basnet.

**Project administration:** Rajindra Napit, Ajit Poudel, Ajay N. Sharma, Amit Basnet.

**Software:** Rajindra Napit, Ajit Poudel, Prajwol Manandhar, Kavya Dhital, Udaya Rajbhandari.

**Supervision:** Jessica S. Schwind, Dibesh B. Karmacharya.

**Validation:** Rajindra Napit, Ajit Poudel, Prajwol Manandhar, Rajesh M. Rajbhandari, Dibesh B. Karmacharya.

**Visualization:** Rajindra Napit, Ajit Poudel, Saman M. Pradhan, Prajwol Manandhar, Suprim Tha, Rajesh M. Rajbhandari.

**Writing – original draft:** Rajindra Napit, Ajit Poudel, Saman M. Pradhan, Prajwol Manandhar, Sajani Ghaju, Jessica S. Schwind, Rajesh M. Rajbhandari, Dibesh B. Karmacharya.

**Writing – review & editing:** Rajindra Napit, Ajit Poudel, Jessica S. Schwind, Dibesh B. Karmacharya.

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
