## [Decision Letter · Decision Letter 0]

19 Sep 2022

PONE-D-22-18860Newcastle disease burden in Nepal and efficacy of Tablet I2 vaccine in commercial and backyard poultry productionPLOS ONE

Dear Dr. Karmacharya, 

Thank you for submitting your manuscript to PLOS ONE. After careful consideration, we feel that it has merit but does not fully meet PLOS ONE’s publication criteria as it currently stands. Therefore, we invite you to submit a revised version of the manuscript that addresses the points raised during the review process.

We look forward to receiving your revised manuscript.

Kind regards,

Shawky M Aboelhadid, PhD

Academic Editor

PLOS ONE

Journal Requirements:

 "Yes- Various components of our effort were supported by PSI (the Netherlands) and InnovationXchange grant from DFAT (Australia)" 

6. Please amend either the abstract on the online submission form (via Edit Submission) or the abstract in the manuscript so that they are identical.

7. Please ensure that you refer to Figure 2 in your text as, if accepted, production will need this reference to link the reader to the figure.

8. We note that Figure 1 in your submission contain [map/satellite] images which may be copyrighted. All PLOS content is published under the Creative Commons Attribution License (CC BY 4.0), which means that the manuscript, images, and Supporting Information files will be freely available online, and any third party is permitted to access, download, copy, distribute, and use these materials in any way, even commercially, with proper attribution. For these reasons, we cannot publish previously copyrighted maps or satellite images created using proprietary data, such as Google software (Google Maps, Street View, and Earth). For more information, see our copyright guidelines: http://journals.plos.org/plosone/s/licenses-and-copyright.

9. We note you have included a table to which you do not refer in the text of your manuscript. Please ensure that you refer to Table 6 and 8 in your text; if accepted, production will need this reference to link the reader to the Table.

Additional Editor Comments:

The manuscript needs English language editing before re-submission. The authors should be care in the revision according to the reviewers comments. 

Reviewers' comments:

Reviewer's Responses to Questions

**Comments to the Author**

1. Is the manuscript technically sound, and do the data support the conclusions?

Reviewer #1: Partly

Reviewer #2: Yes

2. Has the statistical analysis been performed appropriately and rigorously? 

Reviewer #1: N/A

Reviewer #2: Yes

3. Have the authors made all data underlying the findings in their manuscript fully available?

Reviewer #1: Yes

Reviewer #2: Yes

4. Is the manuscript presented in an intelligible fashion and written in standard English?

Reviewer #1: No

Reviewer #2: Yes

5. Review Comments to the Author

Reviewer #1: In this study, serological and molecular surveillance for NDV and IAV among commercial and backyard chickens in Nepal was conducted, Also, vaccine preparation from I-2 NDV strain in two forms was performed and their efficacy was investigated in vivo and in the field study. In addition to these, genetic characterization of NDV GVIIc obtained from sick chicken in Nepal was performed. The work is interesting and huge but some points need to fix before publishing.

Major points:-

1- It is extremely important that this manuscript be reviewed for the English language before it gets resubmitted to correct grammar errors, repetition, and typos.

2- The manuscript needs to reconstruct especially abstract, intoduction, and discussion sections

3- Studies carried out in Nepal to characterize genetically NDV strains and to evaluate vaccine efficacy should be added

4- References list must be revised carefully which references are related and which one should delete.

5- Quality and resolution of figures should be improved

6- Authors should follow the Journal instruction for authors during the preparation of the revised version.

7-The authors should briefly describe in the introduction, the current classification of NDV

Specific points:-

Abstract should be reconstructed and

- line 40: Are these flocks vaccinated? History of vaccination? Is ELISA differentiated between vaccinated and non-vaccinated flocks?

-line 41:The big difference between antibody seroprevalence in commercial and backyard account for which reasons?

-line 41: Genotyped equals strain?

-lne 42: Genotype II or class II

-line 43: Genotype I. Which category is virulent or avirulent? Backyards did not receive a live vaccine?

-lines 43-44:Previous studies did not identify NDV in Nepal until 2021? So, how can conclude the endemicity of NDV?

-line 45: Genotype VIIc belongs to which VII.1.1, VII1.2 or VII.2?

-lne 48: Did you investigate vaccines from this genotype?

Introduction (Background)

-Line 64-65: Cited references (7, 8) does not support this hypothesis?

-Lines 66-71: the NDV classification should be shown according to the updated ICTV classification?

- line 97: studies on papular NDV vaccines in Nepal should be shown.

- lines 101-102: Classification of the vaccine strains virulence including I-2 should be shown. Also, there are studies on the thermo-stability of NDV vaccines and factors that affect vaccine efficacy including I-2 strain and the variability among the different vaccine lots of the same strain. These should be reported here.

- Reference 23 line 109 is an inappropriate reference?

- Lines 110-111: Did you compare the efficiency of these F1 and R2B based vaccines to the current prepared I-2?

- line 113: References took on I-2 only? are there other works on various kinds of NDV vaccines in Nepal? If yes, please highlight it and show its cons.

- Lines 118-123: these look as steps of work not the aim of the study. Please reconstruct to show the main aim of the current study. The authors should clearly state the main objective of the work at the end of the introduction.

Methodology

-Line 129: "the presence of farm personnel" What do you mean by this?

-Line 171: Type of statistic method, program, P value calculation should be described.

-Line 174: "ID screen ELISA for AIV and NDV kits" Could differentiate vaccine strains from virulent strains?

-line 180: "pool of 15 swabs" Is not too much? Too diluted

-lines 235 - 236: " oral and cloacal swabs from two dead chickens" usually swabs collected from alive birds? also, (n=3 ) this stands for what?

-lines 240-241: Could you describe in the detail the preparation process to obtain a tablet and lyophilized forms of live vaccine and references according to which methodology did you do it?

-line 282: "drinking water" why is this route used only for lyophilized formulation only? although in table 3 I can see tablet formulation can be dissolved.

-Lines 288-289: No tablet formulation administrated via drinking water. Why?

-Lines 300 - 305: unclear how did you segregate the birds and how many chickens were sampled?

Results

-Line 347: " All but one farm in ....." as in table 5 one farm in Kailali and one farm in Nawalparasi districts did not receive NDV vaccine.

-Lines 363-364: "All 40 farms vaccinated their stock against at least one of the disease ....." in table 6 only 38 farms not all 40 farms received vaccines, is it right?

-Lines 409 - 410 indicated fragment 2 was amplified for most strains (partial F gene sequences). How long the F gene sequences were obtained to use in Tree drawing? Also, Are there a mutation in amino acids in this region?

-Line 414: How long has the F gene sequence in these 7 NDV (commercial farms=5 and backyard farms=2)? Are there any mutated amino acids observed?

-Line 415: Why the authors did not add the sequences of NDVs from Nepal (as example published in 2000) and I-2 used for vaccine preparation to see a clear picture for the phylogenetic tree?

-Lines 430 - 432: plural used, however, I can see only one sample belongs to Genotype VIIc which is virulent. Figure 10 showed red box that surrounded several NDV strains, what are these and which of them the is target isolate? I can see in the previous publication velogenic strains of NDV from Nepal why the authors did not include here in their analysis to see the virus evolution or genotype variation. Also, the figure is poor quality and resolution and needs to improve.

-Lines 455 - 457: Are the vaccine efficacy differences s 20% and 9% in both ocular and drinking water administration between the two farms in field trials? why?

-Line 460 and line 466: chickens used for field trials are n=2500 in both farms. From where these n=2592 and n=2552 number differences come?

-lines 463-464: is this mortality related to I-2 strain or others?

Discussion

-Discussion section should be one unit without headings or subheadings. Please read the instructions for authors of PlosOne.

-lines 493-495: the meaning is unclear, please reconstruct.

-lines 502-503: please add the number of positive to the total examined beside the percentage.

-Lines 509-510: why the authors did not confirm the pathotypes of NDV, where they sequenced F gene from commercial and backyard samples.

-line 531: authors refer to contact of wild birds with chickens, is this significant?

References

-Several references did not cite well where there is no source for the data as no 3,6, 28, 43, 45 .....etc. Please revise well according to Journal style.

-Several references not related to the research as no 50, 52, 53,57 ...... etc. Please carefully revise your reference list and did not add references just to make a list.

Tables

-Table 1: Total number of farms and samples should be shown. Also, samples from sick birds should be added.

-Table 2: Could you add the fragment position (nt) in the NDV genome and the word fragment in the title row?

-Table 2: the first column means All primers in a single reaction?

-Table 3: Why in vivo trial performed by ocular administration only?

-Table 4: I do not understand this table. Is these birds' separation different than that in figure 4?. Also, you mentioned that a bird is picked up from each section with a total of 12 birds, right? Here too many sampled birds are shown. Could you explain, what is right? In Goldhunga farm, why did not you specify the bird number vaccinated via Ocular in both Tablet and Lyophilized vaccine formulation?

Figures

-Figure 3: please add the nucleotide position for each fragment of the F gene based on NDV genome.

-In Figure 4 legend 70% of chickens received vaccine in DW while in line 290 only 63% of chickens received vaccine in DW. What is the right number?

-Figure 5: I got confused with this figure, what is the significance of sections and quarters? could you simplify your vision by adding group and sub-group numbers to each section and quarter?

-Fig. 9 poor quality can not see, please improve the figure resolution legend please indicate what color (blue, red, and green) mean. How long are the nucleotide sequences

-Figure 12 legend is not descriptive. it should describe the data shown for all groups not only what the authors want to say. in my opinion, both vaccine formulation is similar, is not it? Did you have a significant difference between them?

Reviewer #2: Authors have conducted ND and IAV prevalence study during 2018-19 by collecting samples from 40 commercial and 36 backyard poultry farms from the 10 districts of Nepal. An outbreak was also investigated. Biosamples such as Oral, cloacal and blood samples were pooled and tested for serological and molecular assessment of NDV and IAV. NDV and IAV were detected using multiplex PCR assay. Commercially available ELISA kits were used to detect and quantify ND and Influenza A Nucleoprotein. Three Genotypes I, II and VII c were identified. A thermostable I2 ND vaccine (Ranigoldunga) was developed taking master seed from the University of Queensland, Australia and its Tablet and Lyophilized formulations administered in drinking water and ocular application were evaluated in chicken (Gallus domesticus).

48 (Abstract): The I2 Tablet ND vaccine showed more than 85% efficacy when administered either ocularly or in water, and has stability 50 of 30 days in room temperature.

Tablet form ND vaccine in water is missing in the trial?

141: We collected additional samples from suspected unhealthy or sick birds from commercial (<5 birds) and backyard (< 2 birds) farms prior to our regular random sampling.

Analysis results may be incorporated.

170: Questionnaire used for Biosecurity and biosafety risk assessment in poultry farms is missing?

181: Mention about cloacal swabs in sample processing strategy.

347/358: Description on ELISA as appeared in table 5/6 may be placed in appropriate site.

401: Commercial farms (n= 36) ? Needs correction

503: Justification for variation may be incorporated with reference.

533 & 534: Restructure the sentence to make more meaningful.

561: Replace boiler to broiler

562: More information is required about the vaccine already in use.

General : Additional information on AIV genotyping and Phylogenetic analysis would have added been more informative. Prior to vaccination trial, 1% of the chickens were randomly selected, sampled and screened for NDV and IAV, including some water and feed samples. Results of water and feed samples are missing?

6. PLOS authors have the option to publish the peer review history of their article (what does this mean?). If published, this will include your full peer review and any attached files.

Reviewer #1: No

Reviewer #2: **Yes: **Niranjana Sahoo

---

## [Author Response · Author response to Decision Letter 0]

9 Dec 2022

Authors have conducted ND and IAV prevalence study during 2018-19 by collecting samples from 40 commercial and 36 backyard poultry farms from the 10 districts of Nepal. An outbreak was also investigated. Biosamples such as Oral, cloacal and blood samples were pooled and tested for serological and molecular assessment of NDV and IAV. NDV and IAV were detected using multiplex PCR assay. Commercially available ELISA kits were used to detect and quantify ND and Influenza A Nucleoprotein. Three Genotypes I, II and VII c were identified. A thermostable I2 ND vaccine (Ranigoldunga) was developed taking master seed from the University of Queensland, Australia and its Tablet and Lyophilized formulations administered in drinking water and ocular application were evaluated in chicken (Gallus domesticus). 

48 (Abstract): The I2 Tablet ND vaccine showed more than 85% efficacy when administered either ocularly or in water, and has stability of 30 days in room temperature. [lines 45-46]

Tablet form ND vaccine in water is missing in the trial?

Response: The method of tablet vaccine delivery is intraocular method thus; it was not tested in drinking water. Administration route of the vaccine has been edited. 

141: We collected additional samples from suspected unhealthy or sick birds from commercial (<5 birds) and backyard (< 2 birds) farms prior to our regular random sampling. 

Analysis results may be incorporated. 

Response: This has been removed

170: Questionnaire used for Biosecurity and biosafety risk assessment in poultry farms is missing? 

Response: Questionnaire has been added to supplementary data. 

181: Mention about cloacal swabs in sample processing strategy. 

Response: Cloacal swab sampling procedure explained in method section (159-162). Cloacal swabs were initially collected as back up samples. But upon literature search and optimization processing, oral swabs seemed most effective and only those were processed. This strategy was followed due to resource constraints.

347/358: Description on ELISA as appeared in table 5/6 may be placed in appropriate site. 

Response: Table caption has been edited. ELISA is included in the table to show its correlation with biosecurity measures in the farms. (Table 5/6)

401: Commercial farms (n= 36)? Needs correction 

 Response: Corrected to Backyard farms (n=36) (373)

503: Justification for variation may be incorporated with reference. 

Response: Justification and citation have been added (518-519)

533 & 534: Restructure the sentence to make more meaningful.

 Response: Has been edited and justified by the subsequent sentence in line 547 and 549.

561: Replace boiler to broiler

Response: Boiler has been replaced with broiler (543)

562: More information is required about the vaccine already in use. 

Response: Corrected and citation has been added (550-552)

General : Additional information on AIV genotyping and Phylogenetic analysis would have added been more informative. Prior to vaccination trial, 1% of the chickens were randomly selected, sampled and screened for NDV and IAV, including some water and feed samples. Results of water and feed samples are missing?

Please use the space provided to explain your Responsewers to the questions above. You may also include additional comments for the author, including concerns about dual publication, research ethics, or publication ethics. (Please upload your review as an attachment if it exceeds 20,000 characters)

Reviewer #1: In this study, serological and molecular surveillance for NDV and IAV among commercial and backyard chickens in Nepal was conducted, Also, vaccine preparation from I-2 NDV strain in two forms was performed and their efficacy was investigated in vivo and in the field study. In addition to these, genetic characterization of NDV GVIIc obtained from sick chicken in Nepal was performed. The work is interesting and huge but some points need to fix before publishing.

Major points:-

1- It is extremely important that this manuscript be reviewed for the English language before it gets resubmitted to correct grammar errors, repetition, and typos.

Response – typos have been removed, language and other grammatical errors have been fixed

2- The manuscript needs to reconstruct especially abstract, introduction, and discussion sections

Response – Fixed in manuscript

3- Studies carried out in Nepal to characterize genetically NDV strains and to evaluate vaccine efficacy should be added

Response – studies with molecular characterization of NDV strains have not been published yet, one regarding vaccine efficacy for I-2 has been included [reference #20]

4- References list must be revised carefully which references are related and which one should delete.

Response – Fixed in manuscript

5- Quality and resolution of figures should be improved

Response – fixed in manuscript new figures has been prepared and uploaded.

6- Authors should follow the Journal instruction for authors during the preparation of the revised version.

Response: Those issues has been fixed.

7-The authors should briefly describe in the introduction, the current classification of NDV

Specific points: - Addressed in manuscript

ND is capable of infecting more than 200 species of birds, however, the intensity depends on the host and virus strain. According to Diel et al., NDV is classified into two classes: I and II on the basis of F gene sequence. Class I consists of only one genotype which is avirulent and class II comprises of 15 genotypes (I, II, III, IV, V, VI, VII, VII, IX, X, XI, XII, XIII, XIV and XV). [Genetic diversity of avian paramyxovirus type 1: Proposal for a unified nomenclature and classification system of Newcastle disease virus genotypes https://www.sciencedirect.com/science/article/abs/pii/S1567134812002456?via%3Dihub ] Later, genotypes XVI, XVII, XVIII, XIX, XX and XXI were added to class II which can be found in the latest nomenclature classification system that is used in our study.[Updated unified phylogenetic classification system and revised nomenclature for Newcastle disease virus https://pubmed.ncbi.nlm.nih.gov/31200111/ ] Based on the clinical symptoms, there are five ND virulent pathotype. Class II genotype III NDV were found in Asia, South America, Africa, and Europe. The viruses belonging to this genotype have been identified as velogenic strains. Class II genotype IV NDV were reported in poultry from Africa, Russia, and Europe and in pigeons from Asia. These viruses have been recognized as virulent NDV. Class II genotype V NDVs could be mesogenic or virulent. As per the NDV pandemics, Genotype II, III and IV were responsible for the initial outbreak [Newcastle Disease Viruses Causing Recent Outbreaks Worldwide Show Unexpectedly High Genetic Similarity to Historical Virulent Isolates from the 1940s https://www.ncbi.nlm.nih.gov/pmc/articles/PMC4844730/ ] while the most recent outbreak of NDV was caused by Genotype VII that affected Asia, Africa, Europe and South America. [Insights into Genomic Epidemiology, Evolution, and Transmission Dynamics of Genotype VII of Class II Newcastle Disease Virus in China https://www.mdpi.com/2076-0817/9/10/837 ] 

Abstract should be reconstructed and

- line 40: Are these flocks vaccinated? History of vaccination? Is ELISA differentiated between vaccinated and non-vaccinated flocks?

Response: Vaccination information has been addressed in Table 5 and Table 6 for commercial and backyard farms respectively. 

-line 41: The big difference between antibody seroprevalence in commercial and backyard account for which reasons?

Response: This is addressed in discussion (lines 475-476) – lack of awareness or access to vaccines among backyard poultry farmers can have direct impact on antibody seroprevalence

-line 41: Genotyped equals strain?

Response: The detected NDV was genotyped using sequencing-based approach (F-gene) to be of genotype II in most of the commercial farms. Genotype I was detected in some backyard farms. Fixed in manuscript

-lne 42: Genotype II or class II

Response: Genotype II, fixed in abstract

-line 43: Genotype I. Which category is virulent or avirulent? Backyards did not receive a live vaccine?

Response: Genotype I is usually avirulent strain distributed globally. Backyard chickens in Nepal usually do not receive vaccines, which was the evident from our questionnaire as well [Table 6] 

-lines 43-44: Previous studies did not identify NDV in Nepal until 2021? So, how can conclude the endemicity of NDV?

Response: Our study did not conclusively establish endemicity of NDV but suggested a possibility. Further studies are needed to verify the statement. 

-line 45: Genotype VIIc belongs to which VII.1.1, VII1.2 or VII.2?

Response: Genotype VIIc is a sub-genotype of genotype VII.2 as per citation #47

-line 48: Did you investigate vaccines from this genotype?

Response: Vaccine targeting VII.2 specifically is not available in Nepal. Thus, we could not investigate.

Introduction (Background)

-Line 64-65: Cited references (7, 8) does not support this hypothesis?

Response: Fixed (line 61)

-Lines 66-71: the NDV classification should be shown according to the updated ICTV classification?

Response: Fixed. Corrected in the manuscript (62-63)

- line 97: studies on popular NDV vaccines in Nepal should be shown. 

Response: These have been mentioned in the manuscript (line 98-106)

- lines 101-102: Classification of the vaccine strains virulence including I-2 should be shown. Also, there are studies on the thermo-stability of NDV vaccines and factors that affect vaccine efficacy including I-2 strain and the variability among the different vaccine lots of the same strain. These should be reported here. 

Response: Fixed in manuscript (102-106)

- Reference 23 line 109 is an inappropriate reference? 

Response: This is the correct reference – the statement is supported by Table 5 in the paper (line 98)

- Lines 110-111: Did you compare the efficiency of these F1 and R2B based vaccines to the current prepared I-2? 

Response: We did not compare the strains as studies have already been published with such comparisons. Also, comparing efficiency of the vaccines strains was no the objective of the study. It was to compare the efficiency of our vaccines with endemic strains of NDV.

- line 113: References took on I-2 only? are there other works on various kinds of NDV vaccines in Nepal? If yes, please highlight it and show its cons.

Response: Information on efficacy of other vaccines is limited as mentioned in references 24 and 25 line (100-102). We compared I-2 strains only as it is the thermostable version of the NDV vaccine. Other strains were not thermostable and thus, were mentioned but not the focus of analysis for this study.

- Lines 118-123: these look as steps of work not the aim of the study. Please reconstruct to show the main aim of the current study. The authors should clearly state the main objective of the work at the end of the introduction.

Response: Fixed in manuscript. (107-113)

Methodology- 

-Line 129: "the presence of farm personnel" What do you mean by this?

Response: One of the farm staff was present in sampling site when the sampling procedure was carried out for assisting and supervising. (118)

Response: has been fixed to ‘farm owner or caretaker’ (119)

-Line 171: Type of statistic method, program, P value calculation should be described.

Response: Statistical program has been mentioned in manuscript (line 149)

We did statistical analysis using P-value calculation but these were not significant, hence descriptive analyses were used to provide statistical summary

-Line 174: "ID screen ELISA for AIV and NDV kits" Could differentiate vaccine strains from virulent strains?

Response: These kit alone cannot detect the natural infection from vaccinated ones. The NDVNP kit (BIOVAC used kit) used along with NDV Indirect kit would be able to differentiate natural infection in chickens vaccinated with rHVT-F vaccines only. Similarly, ID Screen® Influenza A Nucleoprotein Indirect used in combination with ID Screen® Influenza H5 Indirect ELISA to detect natural infection in animals vaccinated with recombinant vaccines only (rHVT-H5, RNAm-H5 types). 

-line 180: "pool of 15 swabs" Is not too much? Too diluted 

Response: Pools of 15 swabs represent a single sample from the same farms. In order to minimize the chance of false negative results, the sample collection technique was very thorough collecting the swabs in 500 ul of VTM and vortexing the swabs thoroughly so that all the virus are mixed properly with the VTM and the cold chain was maintained as per lab standards. Normally 1 VTM tubes contains 3 ml for a single sample but we divided this VTM in a sterile environment for 6 samples. So, 15 swabs from a farm is like collecting 5 samples in a 3ml VTM tube. So, it is not as diluted as it may seem (line 159-160)

-lines 235 - 236: " oral and cloacal swabs from two dead chickens" usually swabs collected from alive birds? also, (n=3 ) this stands for what?

Response: The number of samples collected has been fixed. Collection of swab samples varies depending on the type of investigation (can be alive or dead). In this case, recently dead chickens were sampled. Also 2 dead chickens were sampled (n=2) – which has been fixed in manuscript (220)

-lines 240-241: Could you describe in the detail the preparation process to obtain a tablet and lyophilized forms of live vaccine and references according to which methodology did you do it?

Response: Master seed for I2 strain of NDV was obtained from University of Queensland. The lyophilized and tablet vaccines were developed by BIOVAC as mentioned in lines (223-230).

-line 282: "drinking water" why is this route used only for lyophilized formulation only? although in table 3 I can see tablet formulation can be dissolved.

Response: The method of tablet vaccine delivery will be used as intraocular thus; it was not tested in drinking water. (line 272) 

-Lines 288-289: No tablet formulation administrated via drinking water. Why?

Response: Fixed in manuscript (lines 270-272)

-Lines 300 - 305: unclear how did you segregate the birds and how many chickens were sampled?

Response: Segregation of birds and chickens sampled are shown and described in Figure 3 & 4 (new)

Results

-Line 347: " All but one farm in ....." as in table 5 one farm in Kailali and one farm in Nawalparasi districts did not receive NDV vaccine. 

Response: Nawalparasi included in the manuscript (line 315).

-Lines 363-364: "All 40 farms vaccinated their stock against at least one of the disease ....." in table 6 only 38 farms not all 40 farms received vaccines, is it right?

Response: The lines (335-336) has been edited in the manuscript. 

-Lines 409 - 410 indicated fragment 2 was amplified for most strains (partial F gene sequences). How long the F gene sequences were obtained to use in Tree drawing? Also, Are there a mutation in amino acids in this region?

Response: The F gene sequence obtained to build the phylogenetic tree were 521 bp long partial F gene. F0 cleavage sites is present between 110 -117 amino acid positions whereas the sequenced partial F gene was from 160 amino acid positions onward. (lines 381-385)

-Line 414: How long has the F gene sequence in these 7 NDV (commercial farms=5 and backyard farms=2)? Are there any mutated amino acids observed?

Response: The F gene sequence is around 521 bp long. The BCCHT2_2018 (NCBI accession # MZ087886), NWLP4_2018 (NCBI accession # MZ087887), LTP3_2018 (NCBI accession # MZ087888) and BCDNG1_2018 (NCBI accession # MZ087890)], [SNS4_2018 (NCBI accession # MZ087889)] commercial found sequence had mutation on the 176th position Serine (S) into Alanine (A), 192nd Aspargine (N) to Lysine (K), 201st Threonine(T) into Alanine (A). Same as above comments. (lines 392-397)

-Line 415: Why the authors did not add the sequences of NDVs from Nepal (as example published in 2000) and I-2 used for vaccine preparation to see a clear picture for the phylogenetic tree?

Response: There is a phylogenetic tree showing the relation between vaccine strain (I-2) Figure 8 (New) and observed strain in the field.

-Lines 430 - 432: plural used;; however, I can see only one sample belongs to Genotype VII.2 which is virulent. Figure 10 showed red box that surrounded several NDV strains, what are these and which of them the is target isolate? I can see in the previous publication velogenic strains of NDV from Nepal why the authors did not include here in their analysis to see the virus evolution or genotype variation. Also, the figure is poor quality and resolution and needs to improve. 

Response: During the 2021 NDV outbreak we sampled dead chicken whose cause of death was due to NDV and further sequencing the virus led to Genotype VII.2. Two samples were taken but the sequence of only one sample was sequenced. The plural is changed to singular. (407-409)

In the other target isolate obtained from positive NDV screening no virulence cleavage site were sequenced due to which there is no comparison between the target isolates from figure 8(new) which is from screening of chickens all over Nepal. 

New figures 8 & 9 are submitted with high resolution and detailed figure description. (400-404)

-Lines 455 - 457: Are the vaccine efficacy differences 20% and 9% in both ocular and drinking water administration between the two farms in field trials? why? 

Response: Yes, vaccine efficacy differences 20% and 9% in both ocular and drinking water administration between the two farms in field trials respectively. (448-453). The reason has been included in manuscript 

-Line 460 and line 466: chickens used for field trials are n=2500 in both farms. From where these n=2592 and n=2552 number differences come?

Response: This has been fixed. The correct number of chickens used are 2500 in both farms. (444-449)

-lines 463-464: is this mortality related to I-2 strain or others?

Response: This question has been addressed and corresponding data is available in Supplementary tables S8 and S9.

Discussion

-Discussion section should be one unit without headings or subheadings. Please read the instructions for authors of PlosOne.

Response: Fixed in manuscript 

-lines 493-495: the meaning is unclear, please reconstruct.

Response: The section has been removed.

-lines 502-503: please add the number of positive to the total examined beside the percentage.

Response: The actual numbers have been added. (480-483)

-Lines 509-510: why the authors did not confirm the pathotypes of NDV, where they sequenced F gene from commercial and backyard samples.

Response: The sequenced F gene failed to recover F1 fragment completely (F0 cleavage site) so we could not decide the respective pathotype except for outbreak investigation Genotype VII.2. To maintain uniformity, we opted for phylogeny-based analysis (genotyping) and did not perform sequence analysis (mutation analysis).

-line 531: authors refer to contact of wild birds with chickens, is this significant?

Response: Wild birds are known to be involved in transmission and spread of poultry diseases – which has been mentioned in discussion along with relevant citation. (507-510)

References

-Several references did not cite well where there is no source for the data as no 3,6, 28, 43, 45 .....etc. Please revise well according to Journal style.

-Several references not related to the research as no 50, 52, 53,57 ...... etc. Please carefully revise your reference list and did not add references just to make a list.

Response: Citations has been reformatted to Plos one format.

Tables

-Table 1: Total number of farms and samples should be shown. Also, samples from sick birds should be added.

Response: This has been removed as they weren’t used in this study.

-Table 2: Could you add the fragment position (nt) in the NDV genome and the word fragment in the title row?

Response: It has been added on table 2.

-Table 2: the first column meResponse All primers in a single reaction?

Response: The first column doesn’t mean primer in a single reaction. Four different reactions were set for fragment 1, 2, 3 & 4 respectively.

-Table 3: Why in vivo trial performed by ocular administration only?

Response: In vivo trial was performed by ocular administration only as per FAO guidelines for testing vaccines in controlled setting to test the vaccine effectiveness so guidelines were followed strictly. Which was a very early state of the vaccine development, administration via water was later tested in field

-Table 4: I do not understand this table. Is these birds' separation different than that in figure 4?. Also, you mentioned that a bird is picked up from each section with a total of 12 birds, right? Here too many sampled birds are shown. Could you explain, what is right? In Goldhunga farm, why did not you specify the bird number vaccinated via Ocular in both Tablet and Lyophilized vaccine formulation?

Response: figure 3(new) shows the segregation of birds in the farm before vaccination and Table 4 shows the breakdown of birds collected for assessment post-vaccination.

Figures

-Figure 3: please add the nucleotide position for each fragment of the F gene based on NDV genome. 

Response: It has been added. The number has changed in the revision it is now figure 2 (new).

-In Figure 4 legend 70% of chickens received vaccine in DW while in line 290 only 63% of chickens received vaccine in DW. What is the right number? 

Response: The figure depicts 70% were separated for drinking water which was subdivided into two groups one 63% that was actually vaccinated. It is 63% of chickens that received vaccine DW (7% was kept as control: non-vaccinated). It has been mentioned in the figure description.

-Figure 5: I got confused with this figure, what is the significance of sections and quarters? could you simplify your vision by adding group and sub-group numbers to each section and quarter?

Response: The quarters are added to ensure that all chickens have equal chances of getting selected – to minimize selection bias. When collecting chickens, they tend to move around and ones already sampled can get mixed into unsampled group as well. In order to ensure that chicken that has been sampled/vaccinated does not get selected again, the farm was segregated into quarters. 

-Fig. 9 poor quality cannot see, please improve the figure resolution legend please indicate what color (blue, red, and green) mean. How long are the nucleotide sequences

Response: This has been fixed in the manuscript. 

-Figure 11 legend is not descriptive. it should describe the data shown for all groups not only what the authors want to say. in my opinion, both vaccine formulation is similar, is not it? Did you have a significant difference between them?

Response: The legend has been updated in the figure 10 (new) We did not look for significant differences between the groups or controls as we used antibody titre cut-off as sero-positive and sero-negative. Controls clearly were seronegative (less than 800).

Reviewer #2: Authors have conducted ND and IAV prevalence study during 2018-19 by collecting samples from 40 commercial and 36 backyard poultry farms from the 10 districts of Nepal. An outbreak was also investigated. Biosamples such as Oral, cloacal and blood samples were pooled and tested for serological and molecular assessment of NDV and IAV. NDV and IAV were detected using multiplex PCR assay. Commercially available ELISA kits were used to detect and quantify ND and Influenza A Nucleoprotein. Three Genotypes I, II and VII c were identified. A thermostable I2 ND vaccine (Ranigoldunga) was developed taking master seed from the University of Queensland, Australia and its Tablet and Lyophilized formulations administered in drinking water and ocular application were evaluated in chicken (Gallus domesticus).

48 (Abstract): The I2 Tablet ND vaccine showed more than 85% efficacy when administered either ocularly or in water, and has stability 50 of 30 days in room temperature.

Tablet form ND vaccine in water is missing in the trial?

Response: We did not test tablet vaccine on water, as it is intended to be used via ocular route in backyard chickens.

141: We collected additional samples from suspected unhealthy or sick birds from commercial (<5 birds) and backyard (< 2 birds) farms prior to our regular random sampling.

Analysis results may be incorporated.

Response: 

170: Questionnaire used for Biosecurity and biosafety risk assessment in poultry farms is missing?

Response: The questionnaire has been added as supplementary information in separate PDF files.

181: Mention about cloacal swabs in sample processing strategy.

Response: Cloacal swabs were collected but not processed due to budget constraints and also oropharyngeal sample had better sensitivity for detection of NDV and IAV.

347/358: Description on ELISA as appeared in table 5/6 may be placed in appropriate site.

401: Commercial farms (n= 36) ? Needs correction

Response: It has been corrected.

503: Justification for variation may be incorporated with reference.

Response: It has been incorporated.

533 & 534: Restructure the sentence to make more meaningful.

Response: It has been restructured.

561: Replace boiler to broiler

Response: It has been replaced.

562: More information is required about the vaccine already in use.

Response: It has been addressed in background section line 108-116

General: Additional information on AIV genotyping and Phylogenetic analysis would have added been more informative. Prior to vaccination trial, 1% of the chickens were randomly selected, sampled and screened for NDV and IAV, including some water and feed samples. Results of water and feed samples are missing?

6. PLOS authors have the option to publish the peer review history of their article (what does this mean?). If published, this will include your full peer review and any attached files.

Do you want your identity to be public for this peer review? For information about this choice, including consent withdrawal, please see our Privacy Policy.

Reviewer #1: No

Reviewer #2: Yes: Niranjana Sahoo

 

:

---

## [Decision Letter · Decision Letter 1]

26 Dec 2022

PONE-D-22-18860R1Newcastle disease burden in Nepal and efficacy of Tablet I2 vaccine in commercial and backyard poultry productionPLOS ONE

Dear Dr. Dibesh Karmacharya, 

Thank you for submitting your manuscript to PLOS ONE. After careful consideration, we feel that it has merit but does not fully meet PLOS ONE’s publication criteria as it currently stands. Therefore, we invite you to submit a revised version of the manuscript that addresses the points raised during the review process.

ACADEMIC EDITOR: The manuscript needs minor revision depending on reviewers comments. 

We look forward to receiving your revised manuscript.

Kind regards,

Shawky M Aboelhadid, PhD

Academic Editor

PLOS ONE

Journal Requirements:

Reviewers' comments:

Reviewer's Responses to Questions

**Comments to the Author**

1. If the authors have adequately addressed your comments raised in a previous round of review and you feel that this manuscript is now acceptable for publication, you may indicate that here to bypass the “Comments to the Author” section, enter your conflict of interest statement in the “Confidential to Editor” section, and submit your "Accept" recommendation.

Reviewer #1: (No Response)

Reviewer #2: All comments have been addressed

2. Is the manuscript technically sound, and do the data support the conclusions?

Reviewer #1: (No Response)

Reviewer #2: Yes

3. Has the statistical analysis been performed appropriately and rigorously? 

Reviewer #1: (No Response)

Reviewer #2: Yes

4. Have the authors made all data underlying the findings in their manuscript fully available?

Reviewer #1: Yes

Reviewer #2: Yes

5. Is the manuscript presented in an intelligible fashion and written in standard English?

Reviewer #1: Yes

Reviewer #2: Yes

6. Review Comments to the Author

Reviewer #1: Thank you for your responses although I still have some specific points as following

- Concerning your samples' pooling strategy you mention in response, please include it in text to allow research and readership know how it done.

- Concerning your response for using tablet formulation intraocular only, please insert your guide reference support that.

- Although I can find submitted sequences concerning NDV in Nepal before 2021 I did not agree with you that you study is the first report, please revise that.

- These NDV strains in Nepal are interesting, so categorizing them as velogenic or lentogenic is of great important. i recommend you to try F gene cleavage site sequences or MDT as alternative method to sequence to know their virulence, so your trial will be worth.

- Since you use the primers as separated. Please remove the column "Primer mix" in Table 2

- Quality and resolution of figures themselves still need improvements.

Reviewer #2: Authors have responded well to the reviewer's comments.

Authors may continue this study in the same area to unveil endemic nature of the pathogen among the susceptible hosts including wild fauna.

7. PLOS authors have the option to publish the peer review history of their article (what does this mean?). If published, this will include your full peer review and any attached files.

Reviewer #1: No

Reviewer #2: No

While revising your submission, please upload your figure files to the Preflight Analysis and Conversion Engine (PACE) digital diagnostic tool, https://pacev2.apexcovantage.com/. PACE helps ensure that figures meet PLOS requirements. To use PACE, you must first register as a user. Registration is free. Then, login and navigate to the UPLOAD tab, where you will find detailed instructions on how to use the tool. If you encounter any issues or have any questions when using PACE, please email PLOS at figures@plos.org. Please note that Supporting Information files do not need this step.<quillbot-extension-portal></quillbot-extension-portal>

---

## [Author Response · Author response to Decision Letter 1]

2 Jan 2023

6. Review Comments to the Author

Reviewer #1: Thank you for your responses although I still have some specific points as following

- Concerning your samples' pooling strategy you mention in response, please include it in text to allow research and readership know how it done.

Response: Fixed in manuscript (lines 156-164)

- Concerning your response for using tablet formulation intraocular only, please insert your guide reference support that.

Response: The guide reference has been added as a supplementary material

- Although I can find submitted sequences concerning NDV in Nepal before 2021 I did not agree with you that you study is the first report, please revise that.

Response: Fixed in manuscript (line 490)

- These NDV strains in Nepal are interesting, so categorizing them as velogenic or lentogenic is of great important. i recommend you to try F gene cleavage site sequences or MDT as alternative method to sequence to know their virulence, so your trial will be worth.

Response: We have shifted to full length F gene cleavage site-based sequencing. The outbreak investigation of NDV (found to be genotype VII) was carried out with F gene sequencing having cleavage site and is a velogenic strain. However, the old study (nationwide) wide had already been carried out by the time of outbreak and we could not recover full length F gene.

- Since you use the primers as separated. Please remove the column "Primer mix" in Table 2

Response: Fixed in manuscript (line 197)

- Quality and resolution of figures themselves still need improvements.

Reviewer #2: Authors have responded well to the reviewer's comments.

Authors may continue this study in the same area to unveil endemic nature of the pathogen among the susceptible hosts including wild fauna.

7. PLOS authors have the option to publish the peer review history of their article (what does this mean?). If published, this will include your full peer review and any attached files.

Do you want your identity to be public for this peer review? For information about this choice, including consent withdrawal, please see our Privacy Policy.

Reviewer #1: No

Reviewer #2: No

---

## [Editor Report · Decision Letter 2]

6 Jan 2023

Newcastle disease burden in Nepal and efficacy of Tablet I2 vaccine in commercial and backyard poultry production

PONE-D-22-18860R2

Dear Dr. Karmacharya, 

We’re pleased to inform you that your manuscript has been judged scientifically suitable for publication and will be formally accepted for publication once it meets all outstanding technical requirements.

Kind regards,

Shawky M Aboelhadid, PhD

Academic Editor

PLOS ONE

Additional Editor Comments (optional):

Reviewers' comments:

<quillbot-extension-portal></quillbot-extension-portal>

---

## [Editor Report · Acceptance letter]

11 Jan 2023

PONE-D-22-18860R2 

Newcastle disease burden in Nepal and efficacy of Tablet I2 vaccine in commercial and backyard poultry production 

Dear Dr. Karmacharya:

I'm pleased to inform you that your manuscript has been deemed suitable for publication in PLOS ONE. Congratulations! Your manuscript is now with our production department. 

Kind regards, 

on behalf of

Professor Shawky M Aboelhadid 

Academic Editor

PLOS ONE